# BR-SNIS: Bias Reduced Self-Normalized Importance Sampling

**Gabriel Cardoso**

Centre de Mathématiques appliquées,
Ecole polytechnique,
IHU Liryc, fondation Bordeaux Université,
Univ Bordeaux, CRCTB U4045, INSERM,
`gabriel.victorino-cardoso@polytechnique.edu`

**Sergey Samsonov**
HSE University

**Achille Thin**
UMR MIA,
AgroParisTech,

**Eric Moulines**
Centre de Mathématiques appliquées,
Ecole polytechnique,

**Jimmy Olsson**
Department of Mathematics,
KTH Royal Institute of Technology.

## Abstract

Importance Sampling (IS) is a method for approximating expectations under a target distribution using independent samples from a proposal distribution and the associated importance weights. In many applications, the target distribution is known only up to a normalization constant, in which case self-normalized IS (SNIS) can be used. While the use of self-normalization can have a positive effect on the dispersion of the estimator, it introduces bias. In this work, we propose a new method, BR-SNIS, whose complexity is essentially the same as that of SNIS and which significantly reduces bias without increasing the variance. This method is a wrapper in the sense that it uses the same proposal samples and importance weights as SNIS, but makes clever use of iterated sampling–importance resampling (i-SIR) to form a bias-reduced version of the estimator. We furnish the proposed algorithm with rigorous theoretical results, including new bias, variance and high-probability bounds, and these are illustrated by numerical examples.

## 1 Introduction

**Background and previous work:**     *Importance sampling* [17, 1] (IS) is a classical Monte Carlo technique for estimating expectations under some given probability distribution (the *target*) on the basis of a sample of draws from a different distribution (the *proposal*). In the modern era of artificial intelligence and statistical machine learning, characterized by large computational resources and Bayesian inference, IS technologies are enjoying a revival; see, *e.g.*, [37, 23] and [11] for a recent survey. The method is not only relevant to situations where sampling from the target is intractable; it can also be used to achieve variance reduction [24]. When the proposal is dominating the target— in the sense that the support of the latter is contained in the support of the former—unbiased estimation can be achieved by assigning each draw an *importance weight* given by the likelihood ratio between the target and the proposal. In the very common case where the target is known only

36th Conference on Neural Information Processing Systems (NeurIPS 2022).

up to a normalizing constant, consistent estimation can still be achieved by simply normalizing each importance weight by the total weight of the sample; however, since such *self-normalized importance sampling* (SNIS) involves ratios of random variables, the procedure can only be implemented at the cost of bias, which can be significant in some applications.

More precisely, let $(\mathbb{X}, \mathcal{X})$ be some state space and $\pi(\mathrm{d}x) \propto w(x)\lambda(\mathrm{d}x)$ a given target probability distribution, where $w$ and $\lambda$ are a positive weight function and a proposal probability distribution on $(\mathbb{X}, \mathcal{X})$, respectively, such that the normalizing constant $\lambda(w) = \int w(x)\lambda(\mathrm{d}x)$ (this will be our generic notation for Lebesgue integrals) of $\pi$ is finite. The SNIS estimator is given by

$$\Pi_M f(X^{1:M}) = \sum_{i=1}^{M} \omega_M^i f(X^i), \qquad \omega_M^i = w(X^i)/\sum_{\ell=1}^{M} w(X^\ell) \tag{1}$$

where $X^{1:M} = (X^1, \ldots, X^M)$ are independent draws from $\lambda$, and can be used to approximate $\pi(f) = \int f(x)\pi(\mathrm{d}x)$ for any test function $f$ such that $\pi(|f|) < \infty$. The estimator (1) can be calculated without knowledge of the normalizing constant $\lambda(w)$, which is intractable in general.

The SNIS estimator is known to be biased; provided that $\lambda(w^2) < \infty$, the bias and mean-squared error (MSE) of the SNIS estimator (1) over bounded test functions $f$ satisfying $\|f\|_\infty \leq 1$ are given respectively (see [1, Theorem 2.1]) by

$$|\mathbb{E}[\Pi_M f(X^{1:M})] - \pi(f)| \leq (12/M)\kappa[\pi, \lambda], \quad \mathbb{E}[\{\Pi_M f(X^{1:M}) - \pi(f)\}^2] \leq (4/M)\kappa[\pi, \lambda], \tag{2}$$

where $\kappa[\pi, \lambda] = \lambda(w^2)/\lambda^2(w)$. Although IS is primarily intended to approximate integrals in the form $\pi(f)$, it can also be used to generate unweighted samples being approximately distributed according to $\pi$. In this paper, we consider *iterated sampling importance resampling* (i-SIR), proposed in [46]; see [4, 27, 26, 5]. The i-SIR can be seen as an iterative application of the *sampling importance resampling* (SISR) algorithm proposed by [40]; the $k$-th iteration is defined as follows. Given a state $Y_k \in \mathbb{X}$, (i) set $X_{k+1}^1 = Y_k$ and draw $X_{k+1}^{2:N}$ independently from the proposal distribution $\lambda$; (ii) compute, for $i \in \{1, \ldots, N\}$, the normalized importance weights $\omega_{N,k+1}^i = w(X_{k+1}^i)/\sum_{\ell=1}^{N} w(X_{k+1}^\ell)$; (iii) select $Y_{k+1}$ from the set $X_{k+1}^{1:N}$ by choosing $X_{k+1}^i$ with probability $\omega_{N,k+1}^i$. In the following, $Y_{k+1}$ and $X_{k+1}^{1:N}$ will be referred to as the *state* and the *candidate pool*, respectively. Following [46] (see Section 2.1), i-SIR may be viewed (up to an irrelevant permutation of the samples) as a two-stage Gibbs sampler targeting an extended probability distribution $\varphi_N$ on an enlarged state space including the state as well as the candidate pool. As this extended distribution allows $\pi$ as a marginal with respect to the state, one can expect the marginal distribution of the generated states $(Y_k)_{k \in \mathbb{N}}$, forming themselves a Markov chain, to approach the target $\pi$ of interest as $k$ tends to infinity.

**This paper:** In i-SIR, the only function of the candidate pool is to guide the states selected at stage (iii) towards the target. Thus, since all rejected candidates are discarded, the approach results generally in a large waste of computational work. Thus, in the present paper we propose to recycle *all* the generated samples by incorporating all the proposed candidates $X_k^{1:N}$ into the estimator rather than only the selected candidate $Y_k$. We proceed in three steps. First, we show that under the stationary distribution $\varphi_N$ of the process $(Y_k, X_k^{1:N})_{k \in \mathbb{N}}$ generated by i-SIR, the expectation of $\Pi_N f(X_k^{1:N})$ (given by (1)) equals $\pi(f)$ for every valid test function $f$ (see Theorem 2). Second, we establish that since i-SIR is nothing but a systematic-scan Gibbs sampler, the two processes $(X_k^{1:N})_{k \in \mathbb{N}}$ and $(Y_k)_{k \in \mathbb{N}}$ are *interleaving* (see Theorem 5); thus, if $(Y_k)_{k \in \mathbb{N}}$ is uniformly geometrically ergodic, so is $(X_k^{1:N})_{k \in \mathbb{N}}$ with the same mixing rate $\kappa_N$. Third, as the main result of the present paper, we establish a novel $\mathcal{O}(\kappa_N^k/N)$ bound on the bias of the estimator $\Pi_N f(X_k^{1:N})$ (see Theorem 3), where the exponentially diminishing factor $\kappa_N^k$ indicates a drastic bias reduction *vis-à-vis* the standard IS estimator (1) based on i.i.d. samples. As a consequence, approximating $\pi(f)$ by the average of $(\Pi_N f(X_k^{1:N}))_{\ell=k_0+1}^k$, where the "burn-in" period $k_0$ should be chosen proportionally to the mixing time of the process, yields an estimator whose bias can be furnished with a bound which is, roughly, proportional to $\kappa_N^{k_0}$ and inversely proportional to the total number $M = kN$ of samples generated in the algorithm (see Theorem 4). To complete the theoretical analysis of these estimators, we also equip the same with variance bounds. The procedure of recycling, as described above, all the samples generated in the i-SIR and to incorporate, at negligible computational cost, the same into the final estimator, will from now on be referred as BR-SNIS. Finally, we test numerically the proposed estimators and illustrate how a significant bias reduction relatively to the standard i-SIR can be obtained at basically no cost.

To sum up, our contribution is twofold, since we

– propose a new algorithm, BR-SNIS, which makes better use of the available computational resources by recycling the candidate pool generated at each iteration of i-SIR.
– furnish the proposed algorithm with rigorous theoretical results, including novel bias, variance, and high-probability bounds which support our claim that sample recycling may lead to drastic bias reduction without impairing the variance.

## 2 Main results

### 2.1 Statements

The i-SIR algorithm can be interpreted as a systematic-scan two-stage Gibbs sampler, alternately sampling from the full conditions of an extended target $\varphi_N$ on the product space of states and candidate pools. Once the extended target $\varphi_N$ is properly defined, these full conditionals can be retrieved from a dual representation of $\varphi_N$ presented in Theorem 1. In order to define $\varphi_N$, we introduce the Markov kernel (see Appendix A.1 for comments)

$$\mathbf{\Lambda}_N(y, \mathrm{d}x^{1:N}) = N^{-1} \sum_{i=1}^{N} \delta_y(\mathrm{d}x^i) \prod_{j \neq i} \lambda(\mathrm{d}x^j)$$

on $\mathbb{X} \times \mathcal{X}^{\otimes N}$, which describes probabilistically the sampling operation (i) in i-SIR. Using the kernel $\mathbf{\Lambda}_N$ we may now define properly the extended target $\varphi_N$ as the probability law

$$\varphi_N(\mathrm{d}(y, x^{1:N})) = \pi(\mathrm{d}y)\mathbf{\Lambda}_N(y, \mathrm{d}x^{1:N}) = N^{-1} \sum_{i=1}^{N} \pi(\mathrm{d}y)\delta_y(\mathrm{d}x^i) \prod_{j \neq i} \lambda(\mathrm{d}x^j)$$

on $(\mathbb{X}^{N+1}, \mathcal{X}^{\otimes(N+1)})$. Note that since for every $A \in \mathcal{X}$, $\varphi_N(1_{A \times \mathbb{X}^N}) = \pi(A)$, the target $\pi$ coincides with the marginal of $\varphi_N$ with respect to the state. Moreover, it is easily seen that $\mathbf{\Lambda}_N$ provides the conditional distribution, under $\varphi_N$, of the candidate pool given the state. Defining the kernels

$$\Gamma_N(x^{1:N}, \mathrm{d}y) = N^{-1} \sum_{i=1}^{N} w(x^i)\delta_{x^i}(\mathrm{d}y), \quad \Pi_N(x^{1:N}, \mathrm{d}y) = \Gamma_N(x^{1:N}, \mathrm{d}y)/\Gamma_N 1_{\mathbb{X}}(x^{1:N}) \quad (3)$$

on $\mathbb{X} \times \mathcal{X}^{\otimes N}$, the marginal distribution $\boldsymbol{\pi}_N$ of $\varphi_N$ with respect to $x^{1:N}$ is given by

$$\boldsymbol{\pi}_N(\mathrm{d}x^{1:N}) = \lambda(w)^{-1}\Gamma_N 1_{\mathbb{X}}(x^{1:N}) \prod_{j=1}^{N} \lambda(\mathrm{d}x^j). \tag{4}$$

It is interesting to note that the marginal $\boldsymbol{\pi}_N$ has a probability density function, proportional to $\Gamma_N 1_{\mathbb{X}}(x^{1:N}) = \sum_{i=1}^{N} w(x^i)/N$, with respect to the product measure $\lambda^{\otimes N}$. Using (4), we immediately obtain the following result.

**Theorem 1** (duality of extended target). *For every $N \in \mathbb{N}^*$,*

$$\varphi_N(\mathrm{d}(y, x^{1:N})) = \pi(\mathrm{d}y)\mathbf{\Lambda}_N(y, \mathrm{d}x^{1:N}) = \boldsymbol{\pi}_N(\mathrm{d}x^{1:N})\Pi_N(x^{1:N}, \mathrm{d}y). \tag{5}$$

Note that the second identity of the dual representation (5) provides also the conditional distribution, under $\varphi_N$, of the state given the candidates. Consequently, i-SIR is a systematic scan two-stage Gibbs sampler which generates a Markov chain $(X_k, Y_k)_{k \in \mathbb{N}}$ with time-homogeneous Markov kernel

$$\mathbf{P}_N((y_k, x_k^{1:N}), \mathrm{d}(y_{k+1}, x_{k+1}^{1:N})) = \mathbf{\Lambda}_N(y_k, \mathrm{d}x_{k+1}^{1:N})\Pi_N(x_{k+1}^{1:N}, \mathrm{d}y_{k+1})$$

on $\mathbb{X}^{N+1} \times \mathcal{X}^{\otimes(N+1)}$. Note that the law $\mathbf{P}_N(y_k, x_k^{1:N}, \cdot)$ does not depend on $x_k^{1:N}$, which means that only the state $Y_k$ needs to be stored from one iteration to the other. Thus, $(Y_k)_{k \in \mathbb{N}}$ is a Markov chain with Markov transition kernel

$$\mathsf{P}_N(y_k, \mathrm{d}y_{k+1}) = \int \mathbf{\Lambda}_N(y_k, \mathrm{d}x_{k+1}^{1:N})\Pi_N(x_{k+1}^{1:N}, \mathrm{d}y_{k+1}) = \mathbf{\Lambda}_N\Pi_N(y_k, \mathrm{d}y_{k+1}) \tag{6}$$

(where integration is w.r.t. $x_{k+1}^{1:N}$) on $\mathbb{X} \times \mathcal{X}$. The kernel (6) was analyzed in [5]. Given some probability distribution $\boldsymbol{\xi}$ on $(\mathbb{X}^{N+1}, \mathcal{X}^{\otimes(N+1)})$, we denote by $\mathbb{P}_{\boldsymbol{\xi}}$ the law of the canonical Markov chain $(X_k, Y_k)_{k \in \mathbb{N}}$ with kernel $\mathbf{P}_N$ and initial distribution $\boldsymbol{\xi}$. Our first results establishes the unbiasedness of the estimator $\Pi_N f(X^{1:N})$ under $\varphi_N$.

**Theorem 2.** *For every $N \in \mathbb{N}^*$ and $\pi$-integrable function $f$,*

$$\int \Pi_N f(x^{1:N})\boldsymbol{\pi}_N(\mathrm{d}x^{1:N}) = \pi(f).$$

The proof of Theorem 2 is postponed to Appendix A.3. Next, we present theoretical bounds on the discrepancy, in terms of bias, MSE and covariance, between $\Pi_N f(X_k^{1:N})$ and $\pi(f)$, for bounded target functions $f$, when the i-SIR chain is initialized according to an arbitrary distribution $\boldsymbol{\xi}$. We will work under the following assumption.

**A1.** *It holds that $\omega = \|w\|_\infty / \lambda(w) < \infty$.*

Under **A1**, the state and candidate-pool Markov chains $(Y_k)_{k\in\mathbb{N}}$ and $(X^{1:N})_{k\in\mathbb{N}}$ can be shown to be uniformly geometrically ergodic with mixing rate and mixing-time upper bound

$$\kappa_N = (2\omega - 1)/(2\omega + N - 2), \quad \tau_{mix,N} = \lceil -\ln 4 / \ln \kappa_N \rceil, \tag{7}$$

respectively; see Theorem 6 below for details. Here the mixing time $\tau_{mix,N}$ grows logarithmically with the sample size $N$. The exact value of $\tau_{mix,N}$ is likely to be grossly pessimistic, but we conjecture that the logarithmic dependence in the minibatch size holds true. In addition, under **A1** we define the constants

$$
\begin{aligned}
\varsigma^{bias} &= 4(\kappa[\pi,\lambda] + 1 + \omega) \\
\varsigma_i^{mse} &= 4(\kappa[\pi,\lambda]1_{\{0,1\}}(i) + (1+\omega)^2 1_{\{1,2\}}(i)), \quad \varsigma_i^{cov} = \varsigma^{bias}(\varsigma_i^{mse})^{1/2}, \quad i \in \{0,1,2\}.
\end{aligned}
\tag{8}
$$

With these definitions, the following holds true.

**Theorem 3.** *Assume A1. Then for every initial distribution $\boldsymbol{\xi}$ on $(\mathbb{X}^{N+1}, \mathcal{X}^{\otimes(N+1)})$, bounded measurable function $f$ on $(\mathbb{X}, \mathcal{X})$ such that $\|f\|_\infty \leq 1$, $N \geq 2$, and $(k,\ell) \in (\mathbb{N}^*)^2$,*

*(i)* $\left| \mathbb{E}_{\boldsymbol{\xi}}[\Pi_N f(X_k^{1:N})] - \pi(f) \right| \leq \varsigma^{bias}(N-1)^{-1}\kappa_N^{k-1}$,

*(ii)* $\mathbb{E}_{\boldsymbol{\xi}}[\{\Pi_N f(X_k^{1:N}) - \pi(f)\}^2] \leq \sum_{i=0}^{2} \varsigma_i^{mse}(N-1)^{-1-i/2}$,

*(iii)* $\left| \mathbb{E}_{\boldsymbol{\xi}}[\{\Pi_N f(X_k^{1:N}) - \pi(f)\}\{\Pi_N f(X_{k+\ell}^{1:N}) - \pi(f)\}] \right| \leq \kappa_N^{\ell-1} \sum_{i=0}^{2} \varsigma_i^{cov}(N-1)^{-(3-i/2)/2}$,

*where constants are given in* (7) *and* (8).

It is worth noting that the bias decreases inversely with the number of candidates and exponentially with the number of iterations (the mixing time of the chain also depends on $N$). The MSE is also inversely proportional to the number of candidates $N$. In the light of the previous results, it is natural to consider an estimator formed by an average across the IS estimators $(\Pi_N f(X_k^{1:N}))_{k\in\mathbb{N}}$ associated with the candidate pools generated at the different i-SIR iterations. To mitigate the bias, we remove a "burn-in" period whose length $k_0$ should be chosen proportional to the mixing time $\tau_{mix,N}$ of the Markov chain $(Y_k)_{k\in\mathbb{N}}$ (which turns out to coincide with that or the chain $(X_k^{1:N})_{k\in\mathbb{N}}$; see Section 2.2). This yields the estimator

$$\Pi_{(k_0,k),N}(f) = (k - k_0)^{-1} \sum_{\ell=k_0+1}^{k} \Pi_N f(X_\ell^{1:N}) \tag{9}$$

of $\pi(f)$. The total number of samples (generated by the proposal $\lambda$) underlying this estimator is $M = (N-1)k$. Importantly, all the importance weights included in the estimators are obtained as a by-product of the i-SIR schedule; thus, it is, for a given budget of simulations (*i.e.*, under the constraint that $(k-k_0)N$ is constant), possible to compute $\Pi_{(k_0,k),N}(f)$ for different values of $k_0$, $k$ and $N$ with a negligible computational cost. We denote by $\upsilon = (k - k_0)/k$ the ratio of the number of candidate pools used in the estimator to the total number of sampled such pools. Note that this type of estimator was already suggested by [47] and also appears in [42].

Our final main result provides bounds on the bias and the MSE of the estimator (9) as well as a high-probability bound for the same. Define $\zeta^{bias} = 4\tau_{mix,N}\varsigma^{bias}/3$, $\zeta_i^{mse} = \varsigma_{(i+1)\wedge2}^{mse}1_{\{0,2\}}(i) + (8/3)\tau_{mix,N}\varsigma_i^{cov}$, $i \in \{0,1,2\}$, $\zeta^{mse} = \zeta_0^{mse} + \zeta_1^{mse}(N-1)^{-1/4} + \zeta_2^{mse}(N-1)^{-1}$, and $\mathsf{MSE}_M^{is} = (4/M)\kappa[\pi,\lambda]$, see (2).

**Theorem 4.** *Assume A1. Then the following holds true for every initial distribution $\boldsymbol{\xi}$ on $(\mathbb{X}^{N+1}, \mathcal{X}^{\otimes(N+1)})$, bounded measurable function $f$ on $(\mathbb{X}, \mathcal{X})$ such that $\|f\|_\infty \leq 1$, and $N \geq 2$.*

*(i)* $\left| \mathbb{E}_{\boldsymbol{\xi}}[\Pi_{(k_0,k),N}(f)] - \pi(f) \right| \leq \zeta^{bias}(\upsilon M)^{-1}4^{-k_0/\tau_{mix,N}}$

*(ii)* $\mathbb{E}_{\boldsymbol{\xi}}[\{\Pi_{(k_0,k),N}(f) - \pi(f)\}^2] \leq \mathsf{MSE}_{\upsilon M}^{is} + \zeta^{mse}(\upsilon M)^{-1}(N-1)^{-1/2}$

*(iii)* *For every $\delta \in (0,1)$, $|\Pi_{(k_0,k),N}(f) - \pi(f)| \leq \varsigma^{hpd}(\upsilon M)^{-1/2}(\log(4/\delta))^{1/2}$ with probability at least $1 - \delta$, where $\varsigma^{hpd} = 664\omega$.*

**Bootstrap:** As established in Theorem 4, the bias of the BR-SNIS estimator decreases exponentially with the burn-in period $k_0$, leading to potentially significant bias reduction with respect to SNIS. Still, using a large $k_0$ is done at a price of increased overall MSE (mainly through the term $\mathsf{MSE}_{\upsilon M}^{is}$ in Theorem 4(ii), which is directly related to $k_0$ via $\upsilon$). A natural way to reduce the variance is to use

bootstrap. More precisely, we first apply a random permutation to the samples and re-compute BR-SNIS on the basis of the bootstrapped samples. After this, we produce a final estimator by averaging over the bootstrapped BR-SNIS replicates. In most applications, the major computational bottleneck consists of sampling from $\lambda$ and evaluating $w$ and $f$ at the samples; thus, the additional operations that this bootstrap approach entails are computationally cheap. Therefore, in our experiments, we use bootstrap in combination with the choice $k_0 = k - 1$ (in order to minimize the bound in Theorem 4(i)).

## 2.2 Elements of proofs

**Ergodic properties of i-SIR:** The systematic scan two-stage Gibbs sampler is a well-studied MCMC algorithmic structure, and we summarize its most important properties in Theorem 5 below; see [30, 3] and [39, Chapter 9] as well as the references therein. In particular, as shown in [30], the state and candidate-pool Markov chains $(Y_k)_{k \in \mathbb{N}}$ and $(X_k^{1:N})_{k \in \mathbb{N}}$ satisfy a duality property referred to as *interleaving* (Theorem 5(iii)).

**Theorem 5.** *Assume that for every $x \in \mathbb{X}$, $w(x) > 0$, $\lambda(w) < \infty$ and that there exists a set $C \in \mathcal{X}$ such that $\lambda(C) > 0$ and $\sup_{x \in C} w(x)/\lambda(w) < \infty$. Then,*

(i) *the Markov kernel $\mathbf{P}_N$ is Harris recurrent and ergodic with unique invariant distribution $\varphi_N$.*
(ii) *the Markov kernel $\mathsf{P}_N$ is $\pi$-reversible, Harris recurrent and ergodic.*
(iii) *the two Markov chains $(Y_k)_{k \in \mathbb{N}}$ and $(X_k^{1:N})_{k \in \mathbb{N}}$ are conjugate of each other with the interleaving property, i.e., for every initial distribution $\boldsymbol{\xi}$ and $k \in \mathbb{N}$, under $\mathbb{P}_{\boldsymbol{\xi}}$,*
   (a) $X_k^{1:N}$ *and $X_{k+1}^{1:N}$ are conditionally independent given $Y_k$,*
   (b) $Y_k$ *and $Y_{k+1}$ are conditionally independent given $X_{k+1}^{1:N}$;*
   (c) *moreover, under $\mathbb{P}_{\varphi_N}$, $(Y_k, X_{k-1}^{1:N})$ and $(Y_k, X_k^{1:N})$ are identically distributed.*

The ergodic behavior of the i-SIR algorithm has been studied in many works; see [26, 28, 5] in particular. The analysis is particularly simple under the assumption that the importance weight function $w$ is bounded, as imposed by **A**1. Recall that the *total variation-distance* between two probability measures $\xi$ and $\xi'$ on $(\mathbb{X}, \mathcal{X})$ is given by $d_{TV}(\xi, \xi') = \sup_{g:\mathrm{osc}(g) \leq 1}\{\xi(g) - \xi'(g)\}$, where $\mathrm{osc}(g) = \sup_{(x,x') \in \mathbb{X}^2} |g(x) - g(x')|$ denotes the oscillator norm of a measurable function $g$. The following result establishes the uniform geometric ergodicity of the state chain $(Y_k)_{k \in \mathbb{N}}$.

**Theorem 6.** *Assume A1. Then for every $N \geq 2$, $y \in \mathbb{X}$ and $k \in \mathbb{N}$, $d_{TV}(\mathsf{P}_N^k(y, \cdot), \pi) \leq \kappa_N^k$, where $\kappa_N$ is given in* (7).

The proof is given in [28, 5], but we provide it in Appendix A.5 for completeness. For uniformly ergodic Markov chains, it is often more appropriate to work with the mixing time

$$\min\{k \in \mathbb{N} : \sup_{y \in \mathbb{X}} d_{TV}(\mathsf{P}_N^k(y, \cdot), \pi) \leq 1/4\} \leq \tau_{mix,N}$$

(where $\tau_{mix,N}$ is given in (7)), *i.e.*, the number of time steps required for the distribution of the chain to be within a certain total variation distance from its stationary distribution [2, 15]. An interesting consequence of the interleaving property is that if the Markov chain $(Y_k)_{k \in \mathbb{N}}$ is (geometrically) ergodic, then the Markov chain $(X_k^{1:N})_{k \in \mathbb{N}}$ is (geometrically) ergodic as well with the same mixing time; see [39, Corollary 9.14]).

**Bias of the BR-SNIS estimator:** As the BR-SNIS estimator $\Pi_N f(X_k^{1:N})$ (where $\Pi_N$ is defined in (3)) is made up by a ratio of the two unnormalized estimators $\Gamma_N f(X_k^{1:N})$ and $\Gamma_N \mathbb{1}_{\mathbb{X}}(X_k^{1:N})$, a key ingredient in the proof of Theorem 3 is to bound the bias and the $p^{\text{th}}$ order moments of statistics defined as ratios of sums of random variables that are not necessarily independent. The basic idea is to reduce the study of these relations to the analysis of the moments of the numerator and the denominator of these statistics and to exploit their concentration around the respective (conditional and unconditional) means. The main results that we will use in the rest of the paper are summarized in Appendix B.

**Lemma 7.** *For every initial distribution $\boldsymbol{\xi}$ on $(\mathbb{X}^{N+1}, \mathcal{X}^{\otimes(N+1)})$, $k \in \mathbb{N}^*$, and bounded measurable function $f : \mathbb{X} \to \mathbb{R}$, it holds that*

(i) *for every $y \in \mathbb{X}$, $\boldsymbol{\Lambda}_N \Gamma_N f(y) = (1 - 1/N)\lambda(wf) + (1/N)w(y)f(y)$,*
(ii) $\mathbb{E}_{\boldsymbol{\xi}} \left[ \Gamma_N f(X_k^{1:N}) \,\middle|\, Y_{k-1} \right] = \boldsymbol{\Lambda}_N \Gamma_N f(Y_{k-1})$, $\mathbb{P}_{\boldsymbol{\xi}}$-*a.s.,*
(iii) $\mathbb{E}_{\boldsymbol{\xi}} \left[ \{\Gamma_N f(X_k^{1:N}) - \boldsymbol{\Lambda}_N \Gamma_N f(Y_{k-1})\}^2 \,\middle|\, Y_{k-1} \right] = (N-1)/N^2 \lambda(\{wf - \lambda(wf)\}^2)$, $\mathbb{P}_{\boldsymbol{\xi}}$-*a.s.*

We now have all the elements that allow us to determine the first important result of this work, namely the bias and the MSE of the estimator $\Pi_N f(X_k^{1:N})$ of $\pi(f)$.

*Proof of Theorem 3.* We establish the bias bound in (i) and postpone the proof of the bounds on the MSE and the covariance in (ii) and (iii) to the supplement. Define the measure $\xi(A) = \boldsymbol{\xi}(A \times \mathbb{X})$, $A \in \mathcal{X}$, and the kernel $\mathsf{P}_N = \boldsymbol{\Lambda}_N \Pi_N$ on $\mathbb{X} \times \mathcal{X}$. Consequently, $\mathsf{P}_N f(Y_{k-1}) = \mathbb{E}_{\boldsymbol{\xi}}[\Pi_N f(X_k^{1:N}) \mid Y_{k-1}]$ and $\boldsymbol{\Lambda}_N \Gamma_N f(Y_{k-1}) = \mathbb{E}_{\boldsymbol{\xi}}[\Gamma_N f(X_k^{1:N}) \mid Y_{k-1}]$, $\mathbb{P}_{\boldsymbol{\xi}}$-a.s. Since $(Y_k)_{k \in \mathbb{N}}$ is, under $\mathbb{P}_{\boldsymbol{\xi}}$, a Markov chain with initial distribution $\xi$ and Markov kernel $\mathsf{P}_N$ (see (6)), it holds that

$$\mathbb{E}_{\boldsymbol{\xi}}[\Pi_N f(X_k^{1:N})] = \mathbb{E}_{\boldsymbol{\xi}}[\mathsf{P}_N f(Y_{k-1})] = \mathbb{E}_{\boldsymbol{\xi}}[\mathbb{E}_{\boldsymbol{\xi}}[\mathsf{P}_N f(Y_{k-1}) \mid Y_0]] = \xi \mathsf{P}_N^{k-1} \mathsf{P}_N f.$$

Consequently, the proof is concluded by establishing that for every $k \in \mathbb{N}^*$,

$$\left| \xi \mathsf{P}_N^{k-1} \mathsf{P}_N f - \pi(f) \right| \leq \varsigma^{bias} \kappa_N^{k-1} (N-1)^{-1}. \tag{10}$$

On the other hand, since by Theorem 2, $\pi(\mathsf{P}_N f) = \pi(f)$, we may use Theorem 6 to obtain the bound

$$|\xi \mathsf{P}_N^{k-1} \mathsf{P}_N f - \pi(f)| = |\xi \mathsf{P}_N^{k-1} \mathsf{P}_N f - \pi(\mathsf{P}_N f)| \leq \kappa_N^{k-1} \operatorname{osc}(\mathsf{P}_N f).$$

Finally, we establish (10) by bounding $\operatorname{osc}(\mathsf{P}_N f)$. Note that

$$\operatorname{osc}(\mathsf{P}_N f) \leq 2 \left\| \mathsf{P}_N f - \boldsymbol{\Lambda}_N \Gamma_N f / (\boldsymbol{\Lambda}_N \Gamma_N 1_{\mathbb{X}}) \right\|_\infty + 2 \left\| \boldsymbol{\Lambda}_N \Gamma_N f / (\boldsymbol{\Lambda}_N \Gamma_N 1_{\mathbb{X}}) - \pi(f) \right\|_\infty,$$

where, for every $y \in \mathbb{X}$, using Theorem 11,

$$|\mathsf{P}_N f(y) - \boldsymbol{\Lambda}_N \Gamma_N f(y) / \boldsymbol{\Lambda}_N \Gamma_N 1_{\mathbb{X}}(y)|$$
$$\leq \frac{1}{2} \{\boldsymbol{\Lambda}_N \Gamma_N 1_{\mathbb{X}}(y)\}^{-2} \{\boldsymbol{\Lambda}_N[\{\Gamma_N f - \boldsymbol{\Lambda}_N \Gamma_N f(y)\}^2](y) + 3\boldsymbol{\Lambda}_N[\{\Gamma_N 1_{\mathbb{X}} - \boldsymbol{\Lambda}_N \Gamma_N 1_{\mathbb{X}}(y)\}^2](y)\}.$$

Now, since $\boldsymbol{\Lambda}_N \Gamma_N 1_{\mathbb{X}}(y) \geq (1 - 1/N)\lambda(w)$, we get, using Lemma 7,

$$\left\| \mathsf{P}_N f - \frac{\boldsymbol{\Lambda}_N \Gamma_N f}{\boldsymbol{\Lambda}_N \Gamma_N 1_{\mathbb{X}}} \right\|_\infty \leq (2(N-1))^{-1} \{\lambda(w)\}^{-2} \{\lambda(\{wf - \lambda(wf)\}^2) + 3\lambda(\{w - \lambda(w)\}^2)\}$$
$$\leq 2(N-1)^{-1} \lambda(w^2) / (\lambda(w))^2. \tag{11}$$

On the other hand, using the elementary inequality $a/b - c/d = a(d-b)/bd + (a-c)/d$, we get, as $\pi(f) = \lambda(wf)/\lambda(w)$,

$$\frac{\boldsymbol{\Lambda}_N \Gamma_N f(y)}{\boldsymbol{\Lambda}_N \Gamma_N 1_{\mathbb{X}}(y)} - \pi(f) = (1/N) \frac{\boldsymbol{\Lambda}_N \Gamma_N f(y)}{\boldsymbol{\Lambda}_N \Gamma_N 1_{\mathbb{X}}(y)} \{1 - w(y)/\lambda(w)\} + (1/N)\{w(y)f(y) - \lambda(wf)\}/\lambda(w).$$

Finally, the bound (10) is established by noting that

$$\|\boldsymbol{\Lambda}_N \Gamma_N f / (\boldsymbol{\Lambda}_N \Gamma_N 1_{\mathbb{X}}) - \pi(f)\|_\infty \leq 2N^{-1}\{1 + w(y)/\lambda(w)\} \leq 2N^{-1}(1 + \omega). \tag{12}$$

$\square$

## 2.3 Related works

The first use of the IS method, then as a variance reduction technique, dates back to the '50s; see [13, 22] and the references therein. Today, the renewed interest in IS parallels the flurry of activity in the probabilistic ML community and its ever-increasing computational demands; thus, it is impossible to fully present the literature. We therefore limit ourselves to describing results that have inspired our work, and refer the readers to the recent reviews [1, 11] for additional references.

There is clearly a plethora of modern ML applications where the standard SNIS estimator may be substantially improved using the BR-SNIS method. To mention just a selection of examples, SNIS plays a key role for a robust off-policy selection strategy BY [23] (extending [43, 32]), Bayesian problems (see, *e.g.*, [1, Section 3]), Bayesian transfer learning [18, 31], variational autoencoders [9], inference of energy-based models [25], patch-based image restoration [37] and many more. In stochastic-approximation procedures, where a statistical estimator or algorithm is employed repeatedly to produce mean-field estimates, controlling its bias becomes critical [44, 19]. Thus, it is natural to aim at minimizing the bias for a given computational budget, provided that the variance does not explode. For this reason, bias reduction (or unbiasedness) in stochastic simulation has been

the subject of extensive research during the last decades; see [12, 16]. The present paper contributes to this line of research.

Despite long-standing interest in SNIS, there are only few theoretical results. For example, [1, Theorem 2.1] provides bounds on the bias and variance of SNIS, results that we extend to BR-SNIS in Theorem 3. Moreover, [32, Proposition D.3] provides a suboptimal variance bound based on a bound for the second-order moment. This result can be compared to the sophisticated sub-Gaussian concentration bound for BR-SNIS obtained in Theorem 4 (a result that can be obtained for SNIS using the same proof mechanism; see Appendix A.8). Finally, [23] obtains a semi-empirical sub-Gaussian concentration inequality using the Efron-Stein estimate of variance and the Harris inequality.

As an MCMC sampling method, the i-SIR algorithm that has been applied successfully in many situations. It was recently used—under the alternative name *conditional importance sampling*—in [34] for *Markovian score climbing*. In the same work, it is mentioned that it is possible to "Rao-Blackwellize" the gradient of the score using the proposed candidates, which is in line with the recycling argument underpinning the estimator suggested by us, but without theoretical justifications. In its most basic form, the i-SIR algorithm appeared in the pioneering work of [46]. The same idea played a key role in the development of the *particle Gibbs sampler* [4, 5, 35], which extends i-SIR principles to *sequential Monte Carlo methods*. An approach very similar to BR-SNIS can be taken also in this context; however, casting BR-SNIS into the framework of particle Gibbs methods is a non-trivial problem which is the subject of ongoing work.

## 3 Experimental results

In this section we compare numerically the performances of BR-SNIS and SNIS in three different settings: mixture of Gaussians, Bayesian logistic regression and variational autoencoders (VAE). We leave to the supplementary material (Appendix C.1) the detailed numerical verification of the bounds established in Section 2.

**Mixture of Gaussian distributions:** We start with an example where the target distribution $\pi$ is a mixture of two Gaussian distributions of dimension $d = 7$, as shown in Figure 2a. The proposal distribution is a Student distribution with $\nu = 3$ degrees of freedom. The test function is $f = 1_A - 1_B$, where $A$ and $B$ are a $d$-dimensional rectangle intersecting each of the modes of $\pi$ (see Appendix C.1 for precise definitions). We verify the positive effect of bootstrap in Figures 1a and 1b by computing the bias and the MSE over 1000 chains for $N = 129$ for several $k$. The purple, green, and red curves correspond to a number of bootstrap rounds of $1, 21$, and $201$, respectively. We illustrate the decay of the mean Sliced Wasserstein distance (according to [7]) with $k$ for different values of $N$ ($N = 8$ purple, $N = 32$ green, $N = 64$ orange, and $N = 128$ red) in Figure 1c. The decay of the Wasserstein distance is directly linked to the mixing time of the i-SIR kernel (see (7)), and hence allows us to represent the effective mixing time of the chain. Moreover, we represent the theoretical slopes as dashed lines. This illustrates that the effective value of $\tau_{mix,N}$ is smaller than its theoretical bound. The bias and MSE for SNIS with $M = 25600$ are shown in black dashed lines.

We compare the bias (Figure 2b) and MSE (Figure 2c) of BR-SNIS and SNIS for a fixed budget with a total number of $M = 16384$ samples. We run the experiments $10^6$ times; we compute the bias and MSE over batches of $10^4$ replications using the true value of $\pi(f)$ computed above (the boxplots in Figure 2 are therefore obtained over 100 replications). For the algorithm BR-SNIS, we used $N \in \{129, 513\}$, $k_0 = k_{max} - 1$ and $k_{max} = M/(N-1)$ bootstrap rounds. As can be seen from Figure 2b, BR-SNIS significantly reduces bias (by a factor of almost 10) w.r.t. standard SNIS for both configurations, while MSE increases only slightly (at around 20%), as can be seen in Figure 2c. The code used for this experiment is available at [1]. We also show in Appendix C.1 that $k_0 = \lfloor 0.625 k_{max} \rfloor$ can lead to about 3 times less bias w.r.t. standard SNIS while only augmenting the MSE of 10%. We have also compared in BR-SNIS to zero bias estimators based on SNIS such as [33], the results are in shown in Appendix C.1.

**Bayesian Logistic regression:** We consider posterior inference in a Bayesian logistic regression model. Let $\mathcal{D}_{train} = (\mathbf{x}_i, y_i)_{i=1}^T$ be a dataset, where each $\mathbf{x}_i \in \mathbb{R}^d$ is a vector of covariates and $y_i \in \{-1, 1\}$ is a binary response. Let $p(y_i \mid \mathbf{x}_i; \theta) = \{1 + \exp(-y_i \mathbf{x}_i^\top \theta)\}^{-1}$ be the probability of the $i$th observation at $\theta \in \Theta \subseteq \mathbb{R}^d$ and $\pi_0(\mathrm{d}\theta)$ be a prior distribution for $\theta$. The Bayesian posterior is

---

[1]https://github.com/gabrielvc/br_snis

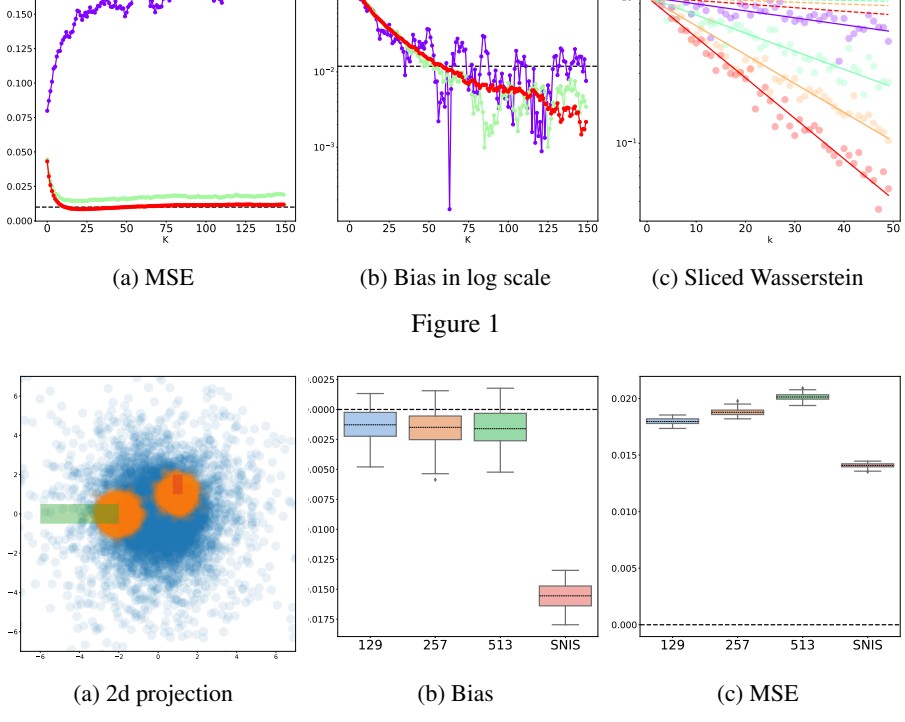

| | | |
|---|---|---|
| (a) MSE | (b) Bias in log scale | (c) Sliced Wasserstein |

Figure 1

| (a) 2d projection | (b) Bias | (c) MSE |
|---|---|---|

Figure 2: Comparison between SNIS and BR-SNIS for the same budget. In each boxplot the dotted line represents the **mean** value of the samples.

given

$$\pi(\mathrm{d}\theta) = \mathrm{Z}^{-1}\pi_0(\mathrm{d}\theta)\exp(\mathcal{L}_T(\theta)), \quad \mathcal{L}_T(\theta) = \sum_{i=1}^{T}\ln p(y_i \mid \mathbf{x}_i; \theta), \quad \mathrm{Z} = \int \exp(\mathcal{L}_T(\theta))\pi_0(\mathrm{d}\theta).$$

For numerical illustration, we use the heart failure clinical records ($d = 13$, $T = 299$), breast cancer detection ($d = 30$, $T = 569$), and Covertype ($d = 55$, $T = 4 \cdot 10^4$) datasets from the UCI machine learning repository. For Covertype, we use Cover type 1 (Spruce/Fir) and Cover type 2 (Lodgepole Pine) classes to define a binary classification problem. As a prior, we use a Gaussian distribution $\mathrm{N}(0, \tau^{-2}\mathbf{I})$ with $\tau^2 = 5 \cdot 10^{-2}$. The importance distribution $\lambda$ is Gaussian with mean and diagonal covariance learned by variational inference; see Appendix C.2 for details. The boxplots for bias in Figure 3 were constructed in the same way as those in Figure 2. We compare two test

| . | CoverType | Breast | Heart |
|---|---|---|---|
| SNIS, M = 32 | 0.0028 +/- 0.0012 | 0.00011 +/- 6.04e-5 | 0.00023 +/- 7.24e-5 |
| BR-SNIS, M= 32 | 0.0014 +/- 0.0003 | 7.9e-5 +/- 5.5e-5 | 0.00012 +/- 6.7e-5 |
| SNIS, M = 512 | 0.0026 +/- 0.0017 | 4.3e-5 +/- 3.3e-5 | 7.8e-5 +/- 6.8e-5 |
| BR-SNIS, M= 512 | 0.0013 +/- 0.0003 | 3.5e-5 +/- 2.2e-5 | 4.9e-5 +/- 5.2e-5 |

Table 1: Comparison of the TV distance between the posteriors (Lower is better).

functions, $f(\theta) = \theta$, corresponding to evaluation of the posterior mean, and $f(\theta) = p(y \mid \mathbf{x}, \theta)$, where $(\mathbf{x}, y) \in \mathcal{D}_{test}$. This last function allows us to compute a TV distance for the predictive distribution. Indeed, in a classification context, one can compute the TV distance between any two predictive distributions $p$ and $\hat{p}$ as

$$d_{TV}(\hat{p}, p) = T^{-1}\sum_{i=1}^{T}\frac{1}{2}\sum_{j=0}^{1}|\hat{p}(y = j \mid \mathbf{x}_i, \mathcal{D}_{train}) - p(y = j \mid \mathbf{x}_i, \mathcal{D}_{train})|,$$

where we compare the predictive distribution $p(y \mid x, \mathcal{D}_{train}) = \int p(y \mid x, \theta)\pi(\theta)\mathrm{d}\theta$ and $\hat{p}$ is the estimation of this quantity, provided in the experiments by SNIS or BR-SNIS. From Figure 3 we can see that for each dataset we have a constant decrease in bias, while the variance increases only slightly. We plot the bias in other components of $\theta$ and provide further numerical details in Appendix C.2.

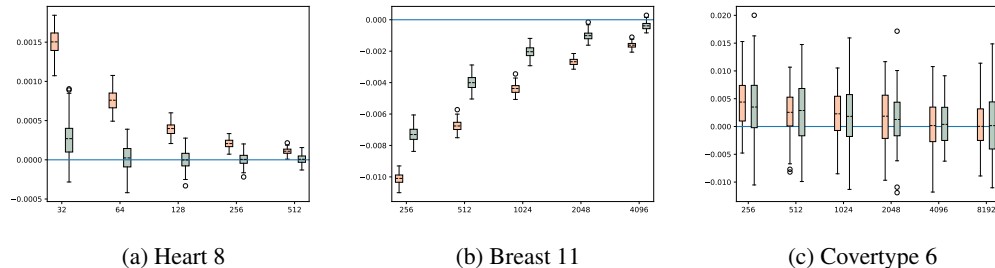

|              | (a) Heart 8 | (b) Breast 11 | (c) Covertype 6 |

Figure 3: Visualization of the distribution for each datasets. Each boxplot is grouped by budget, the left one represent SNIS and the right represent BR-SNIS.

| Latent dimension (d) | VAE | IWAE | BR-IWAE ($k = 8$) |
|---|---|---|---|
| 10 | $-87.40 \pm 0.14$ | $-86.44 \pm 0.10$ | $\mathbf{-86.29 \pm 0.09}$ |
| 20 | $-83.55 \pm 0.10$ | $-81.81 \pm 0.06$ | $\mathbf{-81.66 \pm 0.12}$ |
| 40 | $-82.90 \pm 0.07$ | $-81.05 \pm 0.09$ | $\mathbf{-81.01 \pm 0.05}$ |

Table 2: Comparison of the mean log likelihood over the MNIST validation set (Higher is better).

**Generative Model:** We now extend our methodology to the more complex *deep latent generative models* (DLGM). A DLGM defines a family of probability densities $p_\theta(x)$ over an observation space $x \in \mathbb{R}^P$ by introducing a latent variable $z \in \mathbb{R}^d$, defining the joint density function $p_\theta(x, z)$ (with respect to Lebesgue measure) and aiming to find a parameter $\theta$ maximizing the marginal log-likelihood of the model $p_\theta(x) = \int p_\theta(x, z)\mathrm{d}z$. Under simple technical assumptions, by Fisher's identity,

$$\nabla_\theta \log p_\theta(x) = \int \nabla_\theta \log p_\theta(x, z) p_\theta(z \mid x)\mathrm{d}z, \tag{13}$$

In most cases, the conditional density $p_\theta(z \mid x) = p_\theta(x, z)/p_\theta(x)$ is intractable and can only be sampled. The variational autoencoder [21] is based on the introduction of an additional parameter $\phi$ and a family of variational distributions $q_\phi(z \mid x)$. The joint parameters $\{\theta, \phi\}$ are then inferred by maximizing the *evidence lower bound* (ELBO) defined by

$$\mathcal{L}(\theta, \phi) = \log p_\theta(x) - \mathrm{KL}\big(q_\phi(\cdot \mid x) \parallel p_\theta(\cdot \mid x)\big) \le \log p_\theta(x).$$

This basic setup has been further developed and improved in many directions. Here we consider the *importance weighted autoencoder* (IWAE) [8], which relies on SNIS to design a tighter ELBO on the log-likelihood. The objective of the IWAE is given by

$$\mathcal{L}_M(\theta, \phi) = \int \log \left( M^{-1} \sum_{i=1}^M w_{\theta,\phi,x}(z_i) \right) \prod_{\ell=1}^M q_\phi(z_\ell \mid x)\mathrm{d}z_i,$$

where $w_{\theta,\phi,x}(z) = p_\theta(x, z)/q_\phi(z \mid x)$ denote the importance weights. However, writing, following [8, Eq. (13)],

$$\nabla_\theta \mathcal{L}_M(\theta, \phi) = \int \sum_{i=1}^M \omega_{\theta,\phi,x}^{(i)} \nabla_\theta \log w_{\theta,\phi,x}(z_i) \prod_{\ell=1}^N q_\phi(z_\ell \mid x)\mathrm{d}z_\ell,$$

where $\omega_{\theta,\phi,x}^{(i)} = w_{\theta,\phi,x}(z_i) / \sum_{j=1}^M w_{\theta,\phi,x}(z_j)$ are normalized importance weights, yields an expression of the gradient that corresponds exactly to the biased SNIS approximation of (13). Thus, the optimization problem will suffer from bias. We hence propose to use BR-SNIS for learning IWAE. The proposed algorithm proceeds in two steps, which are repeated during the optimization (details are given in Appendix C.3)

- First, update the parameter $\phi$ as in the IWAE algorithm (using the reparameterization trick and following the methodology of [8]) according to $\phi^{(t+1)} = \phi^{(t)} - \eta\nabla_\phi \mathcal{L}_M(\theta^{(t)}, \phi^{(t)})$.
- Second, update the parameter $\theta$ by estimating (13) using BR-SNIS for $\pi(z) = p_\theta(x, z)$, $f(z) = \nabla_\theta \log p_\theta(x, z)$ and $\lambda(z) = q_\phi(z \mid x)$.

We refer to this model as BR-IWAE. As an illustration, we train the model using the binarized MNIST dataset [41], where $x \in \{0, 1\}^{784}$ are binarized digits images in dimension 784. For both for the encoder $q_\phi$ and the decoder $p_\theta$, we use a convolutional neural network (more details are given in

Appendix C.3). For comparison, we estimate the log-likelihood using the VAE, IWAE and BR-IWAE approaches, and the result is reported in Table 2. All models are run for 100 epochs, using the Adam optimizer [20] and a learning rate of $10^{-4}$. The complete experimental details are given in Appendix C.3.

# 4   Conclusion

In this paper, we have introduced a novel method, BR-SNIS, which improves over SNIS when it comes to producing close to unbiased estimates of expectations taken w.r.t. to distributions known only up to a normalizing constant, a ubiquitous problem in machine learning and statistics. The high performance of BR-SNIS is supported theoretically by non-asymptotic bias, variance and high-probability bounds. We illustrate our method on various examples, which show the practical advantages of BR-SNIS over SNIS. Finally, BR-SNIS is naturally adapted to other IS based methods, for example [45], which use a Hamiltonian (gradient-based) transform [36] as part of the IS proposal. The extension of BR-SNIS to [45] would produce an Hamiltonian based sampler able to recycle all samples, contrarily to other classical Hamiltonian-based methods [36, 14]. BR-SNIS can also be extended to Particle Markov chain Monte Carlo methods such as Particle Gibbs with Ancestor sampling [29].

## Acknowledgement

The work of G. Cardoso is supported through the Investment of the Future grant, ANR-10-IAHU-04, from the government of France through the Agence National de la Recherche. The work of S. Samsonov was supported by SCAI Project-ANR-19-CHIA-0002 and within the framework of the HSE University Basic Research Program. Part of this work has been carried out under the auspice of the Lagrange Center in Mathematics and Computing. The work of J. Olsson is supported by the Swedish Research Council, Grant 2018-05230. The work of E. Moulines is supported by ANR CHIA 002 – SCAI.

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
