# A Proofs

## A.1 i-SIR Algorithm

We analyze a slightly modified version of the i-SIR algorithm, with an extra randomization of the state position. The $k$-th iteration is defined as follows. Given a state $Y_k \in \mathbb{X}$,

   (i) draw $I_{k+1} \in \{1, \ldots, N\}$ uniformly at random and set $X_{k+1}^{I_{k+1}} = Y_k$;
   (ii) draw $X_{k+1}^{1:N \setminus \{I_{k+1}\}}$ independently from the proposal distribution $\lambda$;
   (iii) compute, for $i \in \{1, \ldots, N\}$, the normalized importance weights

$$\omega_{N,k+1}^i = w(X_{k+1}^i) / \sum_{\ell=1}^{N} w(X_{k+1}^\ell);$$

   (iv) select $Y_{k+1}$ from the set $X_{k+1}^{1:N}$ by choosing $X_{k+1}^i$ with probability $\omega_{N,k+1}^i$.

Thus, compared to the simplified i-SIR algorithm given in the introduction, the state is inserted uniformly at random into the list of candidates instead of being inserted at the first position. Of course, this change has no impact as long as we are interested in integrating functions that are permutation invariant with respect to candidates, which is the case throughout our work. Still, this randomization makes the analysis much more transparent.

## A.2 Proof of Theorem 1

We write

$$\varphi_N(\mathrm{d}(y, x_{1:N})) = \frac{1}{N} \sum_{i=1}^{N} \pi(\mathrm{d}y)\delta_y(\mathrm{d}x^i) \prod_{j \neq i} \lambda(\mathrm{d}x^j)$$

$$= \frac{1}{N\lambda(w)} \sum_{i=1}^{N} w(x^i)\lambda(\mathrm{d}x^i)\delta_{x^i}(\mathrm{d}y) \prod_{j \neq i} \lambda(\mathrm{d}x^j)$$

$$= \frac{1}{\lambda(w)} \prod_{j=1}^{N} \lambda(\mathrm{d}x^j)\Gamma_N 1_{\mathbb{X}}(x^{1:N}) \sum_{i=1}^{N} \frac{w(x^i)}{\sum_{\ell=1}^{N} w(x^\ell)}\delta_{x^i}(\mathrm{d}y),$$

where we recognize, and after having recalled definitions (3) and (4) of $\boldsymbol{\pi}_N$ and $\Pi_N$, respectively, the right-hand side as $\boldsymbol{\pi}_N(\mathrm{d}x^{1:N})\Pi_N(x^{1:N}, \mathrm{d}y)$. This completes the proof.

## A.3 Proof of Theorem 2

Using (4) we get

$$\int \boldsymbol{\pi}_N(\mathrm{d}x^{1:N})\Pi_N f(x^{1:N}) = \int \frac{1}{N\lambda(w)} \sum_{\ell=1}^{N} w(x^\ell)\Pi_N f(x^{1:N}) \prod_{j=1}^{N} \lambda(\mathrm{d}x^j)$$

$$= \frac{1}{N\lambda(w)} \int \sum_{i=1}^{N} w(x^i)f(x^i) \prod_{j=1}^{N} \lambda(\mathrm{d}x^j) = \pi(f),$$

and the proof is complete.

## A.4 Proof of Theorem 5

*Proof.* We first check that $\varphi_N$ is an invariant distribution for $\mathbf{P}_N$. For every $A \in \mathcal{X}^{\otimes(N+1)}$, using that $\pi$ is the marginal of $\varphi_N$ with respect to the state and applying Theorem 1 yields

$$\int \varphi_N(\mathrm{d}(y, x^{1:N}))\mathbf{P}_N(y, x^{1:N}, A) = \int \pi(\mathrm{d}y) \iint \boldsymbol{\Lambda}_N(y, \mathrm{d}\bar{x}^{1:N})\Pi_N(\bar{x}^{1:N}, \mathrm{d}\bar{y})1_A(\bar{y}, \bar{x}^{1:N})$$

$$= \iiint \boldsymbol{\pi}_N(\mathrm{d}\bar{x}_{1:N})\Pi_N(\bar{x}^{1:N}, \mathrm{d}y)\Pi_N(\bar{x}^{1:N}, \mathrm{d}\bar{y})1_A(\bar{y}, \bar{x}^{1:N})$$

$$= \varphi_N(A),$$

which establishes invariance. We now show that $\mathsf{P}_N$ is reversible with respect to $\pi$. For this purpose, let $g$ and $h$ be two nonnegative measurable functions and write, using Theorem 1 twice,

$$
\begin{aligned}
\iint \pi(\mathrm{d}y)\mathsf{P}_N(y,\mathrm{d}\bar{y})g(y)h(\bar{y}) &= \int \pi(\mathrm{d}y)\boldsymbol{\Lambda}_N(y,\mathrm{d}x^{1:N})\Pi_N(x^{1:N},\mathrm{d}\bar{y})g(y)h(\bar{y}) \\
&= \int \boldsymbol{\pi}_N(\mathrm{d}x^{1:N})\Pi_N(x^{1:N},\mathrm{d}y)\Pi_N(x^{1:N},\mathrm{d}\bar{y})g(y)h(\bar{y}) \\
&= \int \pi(\mathrm{d}\bar{y})\boldsymbol{\Lambda}_N(\bar{y},\mathrm{d}x^{1:N})\Pi_N(x^{1:N},\mathrm{d}y)g(y)h(\bar{y}) \\
&= \iint \pi(\mathrm{d}\bar{y})\mathsf{P}_N(\bar{y},\mathrm{d}y)g(y)h(\bar{y}).
\end{aligned}
$$

$\square$

## A.5 Proof of Theorem 6

For completeness, we repeat the arguments in [28, 5]. Under **A**1, we have, for $(x,\mathsf{A}) \in \mathbb{X} \times \mathcal{X}$,

$$
\begin{aligned}
\mathsf{P}_N(x,\mathsf{A}) &= \int \delta_x(\mathrm{d}x^1) \sum_{i=1}^{N} \frac{w(x^i)}{\sum_{j=1}^{N} w(x^j)} \mathbb{1}_\mathsf{A}(x^i) \prod_{j=2}^{N} \lambda(\mathrm{d}x^j) \\
&= \int \frac{w(x)}{w(x) + \sum_{j=2}^{N} w(x^j)} \mathbb{1}_\mathsf{A}(x) \prod_{j=2}^{N} \lambda(\mathrm{d}x^j) + \int \sum_{i=2}^{N} \frac{w(x^i)}{w(x) + \sum_{j=2}^{N} w(x^j)} \mathbb{1}_\mathsf{A}(x^i) \prod_{j=2}^{N} \lambda(\mathrm{d}x^j) \\
&\geq \sum_{i=2}^{N} \int \frac{w(x^i)}{w(x) + w(x^i) + \sum_{j=2,j\neq i}^{N} w(x^j)} \mathbb{1}_\mathsf{A}(x^i) \prod_{j=2}^{N} \lambda(\mathrm{d}x^j) \\
&\geq \sum_{i=2}^{N} \int \pi(\mathrm{d}x^i) \mathbb{1}_\mathsf{A}(x^i) \int \frac{\lambda(w)}{w(x) + w(x^i) + \sum_{j=2,j\neq i}^{N} w(x^j)} \prod_{j=2,j\neq i}^{N} \lambda(\mathrm{d}x^j).
\end{aligned}
$$

Finally, since the function $f \colon z \mapsto (z+a)^{-1}$ is convex on $\mathbb{R}_+$ and $a > 0$, we get for $i \in \{2,\ldots,N\}$,

$$
\begin{aligned}
&\int \frac{\lambda(w)}{w(x) + w(x^i) + \sum_{j=2,j\neq i}^{N} w(x^j)} \prod_{j=2,j\neq i}^{N} \lambda(\mathrm{d}x^j) \\
&\geq \frac{\lambda(w)}{\int w(x) + w(x^i) + \sum_{j=2,j\neq i}^{N} w(x^j) \prod_{j=2,j\neq i}^{N} \lambda(\mathrm{d}x^j)} \\
&\geq \frac{1}{w(x)/\lambda(w) + w(x^i)/\lambda(w) + N - 2} \geq \frac{1}{2\omega + N - 2}.
\end{aligned}
$$

We finally obtain the inequality

$$
\mathsf{P}_N(x,\mathsf{A}) \geq \pi(\mathsf{A}) \times \frac{N-1}{2\omega + N - 2} = \epsilon_N \pi(\mathsf{A}). \tag{14}
$$

This means that the whole space $\mathbb{X}$ is $(1, \epsilon_N \pi)$-small (see [10, Definition 9.3.5]). Since $\mathsf{P}_N(x,\cdot)$ and $\pi$ are probability measures, (14) implies

$$
\|\mathsf{P}_N(x,\cdot) - \pi\|_{\mathrm{TV}} = \sup_{\mathsf{A}\in\mathcal{X}} |\mathsf{P}_N(x,\mathsf{A}) - \pi(\mathsf{A})| \leq 1 - \epsilon_N = \kappa_N.
$$

Now the statement follows from [10, Theorem 18.2.4] applied with $m = 1$.

## A.6 Proof of Theorem 3

*Proof of (ii).* Using the identity $(a+b)^2 \leq (1+\epsilon^2)a^2 + (1+\epsilon^{-2})b^2$ we obtain the decomposition $\{\Pi_N f(X_k^{1:N}) - \pi(f)\}^2 \leq (1+(N-1)^{-1/2})\mathrm{I}^{(1)} + (1+(N-1)^{1/2})\mathrm{I}^{(2)}$, with

$$
\begin{aligned}
\mathrm{I}^{(1)} &= \{\Pi_N f(X_k^{1:N}) - a_N(Y_{k-1})/b_N(Y_{k-1})\}^2, \\
\mathrm{I}^{(2)} &= \{a_N(Y_{k-1})/b_N(Y_{k-1}) - \pi(f)\}^2,
\end{aligned}
$$

where $a_N(Y_{k-1}) = \mathbf{\Lambda}_N \Gamma_N f(Y_{k-1})$ and $b_N(Y_{k-1}) = \mathbf{\Lambda}_N \Gamma_N 1_{\mathbb{X}}(Y_{k-1})$.

Using the identity $a/b - c/d = (1/d)[(a/b)(d-b) - (c-a)]$, we obtain

$$\Pi_N f(X_k^{1:N}) - a_N(Y_{k-1})/b_N(Y_{k-1})$$
$$= b_N(Y_{k-1})^{-1} \left[ \Pi_N f(X_k^{1:N})(b_N(Y_{k-1}) - \Gamma_N 1_{\mathbb{X}}(X_k^{1:N})) - (a_N(Y_{k-1}) - \Gamma_N f(X_k^{1:N})) \right].$$

Therefore, using the trivial bound $(a+b)^2 \le 2(a^2+b^2)$, we get

$$\mathrm{I}^{(1)} \le \frac{2}{b_N(Y_{k-1})^2} [\Pi_N f(X_k^{1:N})^2 \{\Gamma_N 1_{\mathbb{X}}(X_k^{1:N}) - b_N(Y_{k-1})\}^2 + \{\Gamma_N f(X_k^{1:N}) - a_N(Y_{k-1})\}^2].$$

Since $\Pi_N f(X_k^{1:N})^2 \le 1$, $\mathbb{P}_{\boldsymbol{\xi}}$-a.s., and $b_N(y) \ge (N-1)/N\lambda(w)$, it holds, $\mathbb{P}_{\boldsymbol{\xi}}$-a.s.,

$$\mathrm{I}^{(1)} \le \frac{2N^2}{(N-1)^2\lambda(w)^2} \left[ \{\Gamma_N 1_{\mathbb{X}}(X_k^{1:N}) - b_N(Y_{k-1})\}^2 + \{\Gamma_N f(X_k^{1:N}) - a_N(Y_{k-1})\}^2 \right].$$

Therefore, using Lemma 7,

$$\mathbb{E}_{\boldsymbol{\xi}}[\{\Pi_N f(X_k^{1:N}) - a_N(Y_{k-1})/b_N(Y_{k-1})\}^2]$$
$$= \mathbb{E}_{\boldsymbol{\xi}}\left[ \mathbb{E}_{\boldsymbol{\xi}}\left[ \{\Pi_N f(X_k^{1:N}) - a_N(Y_{k-1})/b_N(Y_{k-1})\}^2 \,\middle|\, Y_{k-1} \right] \right]$$
$$\le \frac{2N^2}{(N-1)^2\lambda(w)^2} \left[ (N-1)/N^2\lambda(\{w - \lambda(w)\}^2) + (N-1)/N^2\lambda(\{wf - \lambda(wf)\}^2) \right]$$
$$\le 4(N-1)^{-1}\kappa[\pi, \lambda].$$

We turn to $\mathrm{I}^{(2)}$ and note that (12) implies that $\mathrm{I}^{(2)} \le 4N^{-2}(1+\omega)^2$, which completes the proof. $\quad\square$

*Proof of (iii).* Note that

$$\mathrm{I}^{(3)} = \mathbb{E}_{\boldsymbol{\xi}}\left[ \{\Pi_N f(X_k^{1:N}) - \pi(f)\}\{\Pi_N f(X_{k+\ell}^{1:N}) - \pi(f)\} \right]$$
$$= \mathbb{E}_{\boldsymbol{\xi}}[\{\Pi_N f(X_k^{1:N}) - \pi(f)\}\mathbb{E}_{\boldsymbol{\xi}}\left[ \Pi_N f(X_{k+\ell}^{1:N}) - \pi(f) \,\middle|\, Y_{k+\ell-1} \right]].$$

As $\mathbb{E}_{\boldsymbol{\xi}}\left[ \Pi_N f(X_{k+\ell}^{1:N}) \,\middle|\, Y_{k+\ell-1} \right] = \Phi_N(Y_{k+\ell-1})$ $\mathbb{P}_{\boldsymbol{\xi}}$-a.s., it holds that

$$\mathrm{I}^{(3)} = \mathbb{E}_{\boldsymbol{\xi}}[\{\Pi_N f(X_k^{1:N}) - \pi(f)\}\{\Phi_N(Y_{k+\ell-1}) - \pi(f)\}]$$
$$= \mathbb{E}_{\boldsymbol{\xi}}[\{\Pi_N f(X_k^{1:N}) - \pi(f)\}\{\mathbb{E}_{\boldsymbol{\xi}}\left[ \Phi_N(Y_{k+\ell-1}) \,\middle|\, Y_k \right] - \pi(f)\}].$$

By the Markov property,

$$\mathbb{E}_{\boldsymbol{\xi}}\left[ \Phi_N(Y_{k+\ell-1}) \,\middle|\, Y_k \right] = \mathsf{P}_N^{\ell-1}\Phi_N(Y_k) = \delta_{Y_k}\mathsf{P}_N^{\ell-1}\Phi_N, \quad \mathbb{P}_{\boldsymbol{\xi}}\text{-a.s.},$$

which, combined with (10), implies that

$$\|\mathsf{P}_N^{\ell-1}\Phi_N - \pi(f)\|_\infty \le \varsigma^{bias}(N-1)^{-1}\kappa_N^{\ell-1}.$$

Combining the results above, we finally establish that

$$|\mathrm{I}^{(3)}| \le \varsigma^{bias}(N-1)^{-1}\kappa_N^{\ell-1}\mathbb{E}_{\boldsymbol{\xi}}[\{\Pi_N f(X_k^{1:N}) - \pi(f)\}^2]^{1/2}$$
$$\le \varsigma^{bias}(N-1)^{-1}\kappa_N^{\ell-1} \left( \sum_{i=0}^2 \varsigma_i^{mse}(N-1)^{-1-i/2} \right)^{1/2}.$$

$\quad\square$

## A.7 Proof of Theorem 4

We first consider the bias term, which can be bounded according to

$$\left| \mathbb{E}_{\boldsymbol{\xi}}[\Pi_{(K_0,K),N}(f)] - \pi(f) \right| \le (K-K_0)^{-1} \sum_{\ell=K_0+1}^{K} \left| \mathbb{E}_{\boldsymbol{\xi}}[\Pi_N f(X_\ell^{1:N})] - \pi(f) \right|$$
$$\le (K-K_0)^{-1}(N-1)^{-1}\varsigma^{bias} \sum_{\ell=K_0+1}^{K} \kappa_N^{\ell-1}.$$

Thus, the claimed bias bound can be established by noting that

$$\sum_{\ell=K_0+1}^{K} \kappa_N^{\ell-1} \leq \frac{\kappa_N^{K_0}}{1-\kappa_N} \leq \frac{4\tau_{mix,N}(1/4)^{K_0/\tau_{mix,N}}}{3}.$$

We now turn to the MSE, and make the decomposition

$$\mathbb{E}_{\boldsymbol{\xi}}[\{\Pi_{(K_0,K),N}(f) - \pi(f)\}^2] \leq (K-K_0)^{-2} \left( \sum_{\ell=K_0+1}^{K} \mathbb{E}_{\boldsymbol{\xi}}[\Pi_N f(X_\ell^{1:N})] - \pi(f) \right)^2$$

$$+ 2 \sum_{\ell=K_0+1}^{K} \sum_{j=\ell+1}^{K} \mathbb{E}_{\boldsymbol{\xi}}[\{\Pi_N f(X_\ell^{1:N}) - \pi(f)\}\{\Pi_N f(X_j^{1:N}) - \pi(f)\}].$$

Using the MSE bound in Theorem 11, we obtain that

$$\sum_{\ell=K_0+1}^{K} \mathbb{E}_{\boldsymbol{\xi}}[\{\Pi_N f(X_\ell^{1:N}) - \pi(f)\}^2] \leq (K-K_0)(N-1)^{-1} \sum_{i=0}^{2} \varsigma_i^{mse}(N-1)^{-i/2}.$$

In addition, the covariance bound of Theorem 11 yields

$$\sum_{\ell=K_0+1}^{K} \sum_{j=\ell+1}^{K} \mathbb{E}_{\boldsymbol{\xi}}[\{\Pi_N f(X_\ell^{1:N}) - \pi(f)\}\{\Pi_N f(X_j^{1:N}) - \pi(f)\}]$$

$$\leq \sum_{i=0}^{2} \varsigma_i^{cov}(N-1)^{-(3-i/2)/2} \left( \sum_{\ell=K_0+1}^{K} \sum_{j=\ell+1}^{K} \kappa_N^{(j-\ell)-1} \right).$$

As $\sum_{\ell=K_0+1}^{K} \sum_{j=\ell+1}^{K} \kappa_N^{(j-\ell)-1} \leq (K-K_0)(4/3)\tau_{mix,N}$, we may write

$$\mathbb{E}_{\boldsymbol{\xi}}[(\Pi_{(K_0,K),N}(f) - \pi(f))^2] \leq ((K-K_0)(N-1))^{-1} \left( \sum_{i=0}^{2} \varsigma_i^{mse}(N-1)^{-i/2} \right)$$

$$+ (8/3)(K-K_0)^{-1}(N-1)^{-3/2} \left( \sum_{i=0}^{2} \varsigma_i^{cov}(N-1)^{-i/4} \right),$$

and the MSE bound may now be established by noting that $(K-K_0)(N-1) = \upsilon M$.

Establishing the high-probability bound requires more complex derivations. More precisely, we will apply the decomposition

$$\Pi_{(K_0,K),N}(f) - \pi(f) = (K-K_0)^{-1} \sum_{k=K_0+1}^{K} \Pi_N f(X_k^{1:N}) - \Phi_N(Y_{k-1})$$

$$+ (K-K_0)^{-1} \sum_{k=K_0+1}^{K-1} \Phi_N(Y_{k-1}) - \pi(\Phi_N),$$

where we used that $\pi(f) = \pi(\Phi_N)$. Therefore, for every $t \geq 0$ it holds that

$$\mathbb{P}_{\boldsymbol{\xi}}(|\Pi_{(K_0,K),N}(f) - \pi(f)| \geq t) \leq \mathbb{P}_{\boldsymbol{\xi}} \left( (K-K_0)^{-1} \left| \sum_{k=K_0+1}^{K} \Pi_N f(X_k^{1:N}) - \Phi_N(Y_{k-1}) \right| \geq t/2 \right)$$

$$+ \mathbb{P}_{\boldsymbol{\xi}} \left( (K-K_0)^{-1} \left| \sum_{k=K_0+1}^{K-1} \Phi_N(Y_{k-1}) - \pi(\Phi_N) \right| \geq t/2 \right).$$

We will show that for all $t > 0$, and for some absolute constants $\zeta^{(1)}$ and $\zeta^{(2)}$,

$$\mathrm{I}^{(1)} = \mathbb{P}_{\boldsymbol{\xi}}\left((K-K_0)^{-1}\left|\sum_{k=K_0+1}^{K}\Pi_N f(X_k^{1:N}) - \Phi_N(Y_{k-1})\right| \geq t\right) \leq 2\exp(-t^2 \upsilon M/(4\zeta^{(1)})),$$

$$\mathrm{I}^{(2)} = \mathbb{P}_{\boldsymbol{\xi}}\left((K-K_0)^{-1}\left|\sum_{k=K_0+1}^{K-1}\Phi_N(Y_{k-1}) - \pi(\Phi_N)\right| \geq t\right)$$

$$\leq 2\exp(-t^2\zeta^{(2)}(K-K_0)(N-1)^2/\tau_{mix,N}).$$

We first consider $\mathrm{I}^{(1)}$ and note that

$$\mathrm{I}^{(1)} = \mathbb{E}_{\boldsymbol{\xi}}\left[\mathbb{P}_{\boldsymbol{\xi}}\left((K-K_0)^{-1}\left|\sum_{k=K_0+1}^{K}\Pi_N f(X_k^{1:N}) - \Phi_N(Y_{k-1})\right| \geq t \mid Y_{K_0:K-1}\right)\right].$$

By Theorem 5, the random elements $(X_k^{1:N})_{k=K_0+1}^{K}$ are independent conditionally to $(Y_k)_{k=K_0}^{K-1}$. Thus, using the generalized Hoeffding inequality (see [48, Theorem 2.6.2] or [49, Proposition 2.1]) we get, with $\Delta_{N,k} = \Pi_N f(X_k^{1:N}) - \Phi_N(Y_{k-1})$, that, $\mathbb{P}_{\boldsymbol{\xi}}$-a.s.,

$$\mathbb{P}_{\boldsymbol{\xi}}\left((K-K_0)^{-1}\left|\sum_{k=K_0+1}^{K}\Delta_{N,k}\right| \geq t \mid Y_{K_0:K-1}\right) \leq 2\exp\left(-\frac{t^2(K-K_0)^2}{4\sum_{k=K_0+1}^{K}\|\Delta_{N,k}\|_{\psi_2,Y_k}^2}\right),$$

where $\psi_2 : x \mapsto \exp(x^2) - 1$ and

$$\|\Delta_{N,k}\|_{\psi_2,Y_{k-1}} = \inf\{\lambda > 0 : \mathbb{E}_{\boldsymbol{\xi}}\left[\psi_2(|\Delta_{N,k}|/\lambda) \mid Y_{k-1}\right] \leq 1\}.$$

In order to bound $\|\Delta_{N,k}\|_{\psi_2,Y_{k-1}}$ we use the decomposition $\Delta_{N,k} = \Delta_{N,k}^{(1)} + \Delta_{N,k}^{(2)}$, where

$$\Delta_{N,k}^{(1)} = \frac{\Gamma_N f(X_k^{1:N})}{\Gamma_N \mathbb{1}_{\mathbb{X}}(X_k^{1:N})} - \frac{a_N(Y_{k-1})}{b_N(Y_{k-1})},$$

$$\Delta_{N,k}^{(2)} = \frac{a_N(Y_{k-1})}{b_N(Y_{k-1})} - \Phi_N(Y_{k-1}),$$

combined with Lemma 9 with $\phi = \chi = \psi_2$ and [48, Proposition 2.6.1]. By (11) and by [48, Equation 2.17] it holds that, $\mathbb{P}_{\boldsymbol{\xi}}$-a.s.,

$$\|\Delta_{N,k}^{(2)}\|_{\psi_2,Y_{k-1}} \leq 2(\log 2)^{-1/2}(N-1)^{-1}\kappa[\lambda,\pi].$$

Using Lemma 9 with $\phi = \chi = \psi_2$ and the fact that $b_N(y) \geq (1 - 1/N)\lambda(w)$ we obtain, $\mathbb{P}_{\boldsymbol{\xi}}$-a.s.,

$$\|\Delta_{N,k}^{(1)}\|_{\psi_2,Y_{k-1}}$$
$$\leq \frac{2}{(1-1/N)\lambda(w)}\left(\|\Gamma_N f(X_k^{1:N}) - a_N(Y_{k-1})\|_{\psi_2,Y_{k-1}} + 2\|\Gamma_N \mathbb{1}_{\mathbb{X}}(X_k^{1:N}) - b_N(Y_{k-1})\|_{\psi_2,Y_{k-1}}\right).$$

Furthermore, using [48, Proposition 2.6.1, Eq 2.17] we get, $\mathbb{P}_{\boldsymbol{\xi}}$-a.s.,

$$\|\Gamma_N f(X_{k-1}^{1:N}) - a_N(Y_{k-1})\|_{\psi_2,Y_{k-1}}^2$$
$$\leq (64\mathrm{e}/\log 2)N^{-1}\left\|w(X_k^1)f(X_k^1) - \mathbb{E}_{\boldsymbol{\xi}}\left[w(X_k^1)f(X_k^1) \mid Y_{k-1}\right]\right\|_{\psi_2,Y_{k-1}}^2,$$
$$\leq (256\mathrm{e}/(\log 2)^2)N^{-1}\|w\|_\infty^2.$$

The same bound applies to $\|\Gamma_N \mathbb{1}_{\mathbb{X}}(X_k^{1:N}) - b_N(Y_{k-1})\|_{\psi_2,Y_{k-1}}^2$, and we may write

$$\|\Delta_{N,k}^{(1)}\|_{\psi_2,Y_{k-1}} \leq 96\mathrm{e}^{1/2}(\log 2)^{-1}(N-1)^{-1/2}\omega.$$

We can now finalize the bound on $\mathrm{I}^{(1)}$ by writing

$$\|\Delta_{N,k}\|_{\psi_2,Y_{k-1}}^2 \leq 2(\|\Delta_{N,k}^{(1)}\|_{\psi_2,Y_{k-1}}^2 + \|\Delta_{N,k}^{(2)}\|_{\psi_2,Y_{k-1}}^2)$$
$$\leq (N-1)^{-1}(\zeta^{(1,1)}\omega^2 + \zeta^{(1,2)}\kappa[\lambda,\pi]^2(N-1)^{-1}),$$

where $\zeta^{(1,1)} = 18432\mathrm{e}(\log 2)^{-2}$ and $\zeta^{(1,2)} = 8(\log 2)^{-1}$ are universal constants, which implies that

$$\|\Delta_{N,k}\|_{\psi_2,Y_{k-1}}^2 \le \zeta^{(1)}(N-1)^{-1},$$

with $\zeta^{(1)} = 1.1 \cdot 10^5 \omega^2$. This finally yields that $\mathrm{I}^{(1)} \le 2\exp(-t^2 \upsilon M/4\zeta^{(1)})$.

We treat $\mathrm{I}^{(2)}$ using Lemma 12 with $g_i = \Phi_N(Y_{K_0+i-1}) - \pi(\Phi_N)$. As $\|g_i\|_\infty \le \mathrm{osc}(\Phi_N) \le (N-1)^{-1}\varsigma^{bias}$, we obtain

$$\mathrm{I}^{(2)} \le 2\exp\left(-t^2 \zeta^{(2)}(K-K_0)(N-1)^2/\tau_{mix,N}\right),$$

where $\zeta^{(2)} = 2/(3\varsigma^{bias})^2$. Finally, we obtain

$$\mathbb{P}_{\boldsymbol{\xi}}(|\Pi_{(K_0,K),N}(f) - \pi(f)| \ge t)$$
$$\le 2\exp\left(-t^2\upsilon M/4\zeta^{(1)}\right)\left[1 + \exp\left(-t^2\upsilon M\{\zeta^{(2)}(N-1)/\tau_{mix,N} - (4\zeta_I)^{-1}\}\right)\right].$$

We conclude by noting that for every $\delta \in (0,1)$ and $N-1 \ge \tau_{mix,N}(4\zeta^{(1)}\zeta^{(2)})^{-1}$ it holds that

$$\mathbb{P}_{\boldsymbol{\xi}}(|\Pi_{(K_0,K),N}(f) - \pi(f)| \ge t) \le 4\exp\left(-t^2\upsilon M/4\zeta^{(1)}\right) \le \delta$$

for all $t \ge 2\zeta_I^{1/2}(\upsilon M)^{-1/2}\log(4/\delta)^{1/2}$. Letting $\varsigma^{hpd} = 2\zeta_I^{1/2}$ concludes the proof.

### A.8 High-probability inequality for SNIS

**Theorem 8.** *Assume that $\omega = \|w\|_\infty/\lambda(w) < \infty$. For all bounded measurable functions $f$ on $(\mathbb{X},\mathcal{X})$ such that $\|f\|_\infty \le 1$, it holds that for every $M \in \mathbb{N}^*$ and $\delta \in (0,1)$,*

$$|\widehat{\pi}_M(f) - \pi(f)| \le 12\omega(M\log 2)^{-1/2}\log(2/\delta)^{1/2}$$

*with probability larger than $1 - \delta$.*

*Proof.* Let $\alpha_M = M^{-1}\sum_{i=1}^M w(X^i)f(X^i)$, $\beta_M = M^{-1}\sum_{i=1}^M w(X^i)$, $a = \mathbb{E}[\alpha_M] = \lambda(wf)$, and $b = \mathbb{E}[\beta_M] = \lambda(w)$. Note that $\widehat{\pi}_M(f) = \alpha_M/\beta_M$ and $\pi(f) = a/b$. Using Lemma 9 with $\phi$ and $\chi$ equal to the mapping $x \mapsto \exp(x^2) - 1$ we obtain that

$$\|\widehat{\pi}_M(f) - \pi(f)\|_{\psi_2} \le 2\lambda(w)^{-1}\left(\|\alpha_M - a\|_{\psi_2} + 2\|\beta_M - b\|_{\psi_2}\right).$$

Moreover, using [48, Eq 2.17] yields, $\mathbb{P}_{\boldsymbol{\xi}}$-a.s.,

$$\|\alpha_M - a\|_{\psi_2}^2 \le M^{-1}\|w(X^i)f(X^i) - \lambda(wf)\|_{\psi_2}^2 \le 4(M\log 2)^{-1}\|w\|_\infty^2.$$

In the same way, $\|\beta_M - b\|_{\psi_2}^2 \le 4(M\log 2)^{-1}\|w\|_\infty^2$. Therefore, we may conclude that

$$\|\widehat{\pi}_M(f) - \pi(f)\|_{\psi_2}^2 \le (12\omega)^2(M\log 2)^{-1}.$$

Combining the previous bound with [48, Proposition 2.5.2] provides

$$\mathbb{P}(|\widehat{\pi}_M(f) - \pi(f)| \ge t) \le 2\exp(-t^2\zeta^{snis}M),$$

where $\zeta^{snis} = (12\omega)^{-2}\log 2$. The high-probability inequality of the theorem follows directly. $\square$

## B Moments and high-probability bounds for ratio statistics

Let $(U_i, V_i)_{i\in\{1,\dots,n\}}$ be (possibly dependent) random variables defined on some probability space $(\Omega, \mathcal{F}, \mathbb{P})$. Assume that $U_i \ge 0$ $\mathbb{P}$-a.s. Moreover, let $\alpha_n = n^{-1}\sum_{i=1}^n U_iV_i$, $\beta_n = n^{-1}\sum_{i=1}^n U_i$, and $\rho_n = \alpha_n/\beta_n$ as well as $a = \mathbb{E}[\alpha_n]$, $b = \mathbb{E}[\beta_n]$, and $r = a/b$.

A continuous, even, convex function $\phi : \mathbb{R}^+ \to [0,+\infty]$ is a Young function if $\phi$ is monotonically increasing for $x > 0$, $\phi(0) = 0$, $\lim_{x\to\infty}\phi(x)/x = \infty$, and $\lim_{x\to 0^+}\phi(x)/x = 0$. We denote by $\phi^*$ the Fenchel-Legendre conjugate of $\phi$. Let $X$ be a random variable and $\phi$ a Young function. Then the *Orlicz norm* of $X$ is

$$\|X\|_\phi = \inf\{\lambda > 0 : \mathbb{E}[\phi(|X|/\lambda)] \le 1\},$$

with the convention that $\inf \emptyset = \infty$. The Orlicz space $\mathcal{L}_\phi(\Omega)$ of random variables is the family of equivalence classes of random variables $X$ such that $\|X\|_\phi < \infty$. Here $\mathcal{L}_\phi(\Omega)$ is a Banach space. If $\phi_p(x) = |x|^p$ for $p \ge 1$, then $\mathcal{L}_\phi(\Omega) = \mathcal{L}^p(\Omega)$ and we denote $\|\cdot\|_p = \|\cdot\|_{\phi_p}$. If $X \in \mathcal{L}_\phi(\Omega)$, then, for every $x > 0$,

$$\mathbb{P}(|X| \ge x) \le 1/\phi(x/\|X\|_\phi) \quad \text{and} \quad \|1_{\{|X|\ge x\}}\|_\phi = 1/\phi^{-1}(1/\mathbb{P}(|X| \ge x)).$$

**Lemma 9.** *Let $\phi$ and $\chi$ be Young functions. If $\max_i \|V_i\|_\infty \le c|r|$, then*

$$\|\rho_n - r\|_\phi/|r| \le 2\|\alpha_n - a\|_\phi/b + 2\|\beta_n - b\|_\phi/b + c/\{(\phi^{-1} \circ \chi)(b/2\|(\beta_n - b)_-\|_\chi)\}.$$

*Proof.* We decompose the computation in two parts: first, when $\beta_n > b/2$, we have

$$|\rho_n - r| = \left| \frac{\alpha_n - a}{\beta_n} + a\left(\frac{1}{\beta_n} - \frac{1}{b}\right) \right| \le \frac{|\alpha_n - a|}{b/2} + \frac{|a||\beta_n - b|}{(b/2)b} = \frac{2|\alpha_n - a|}{b} + \frac{2|r||\beta_n - b|}{b}.$$

Then, when $\beta_n \le b/2$,

$$|\rho_n - r| \le |\rho_n| + |r| \le |\rho_n| + \frac{2|r||\beta_n - b|}{b} \le \max_i |V_i| + \frac{2|r||\beta_n - b|}{b},$$

where the second inequality follows from $|\beta_n - b| \ge b/2$. Combining the two previous inequalities yields

$$|\rho_n - r| \le \frac{2|\alpha_n - a|}{b} + \frac{2|r||\beta_n - b|}{b} + \max_i |V_i| 1_{\{\beta_n \le b/2\}}.$$

Recall that if $|X| \le |Y|$ $\mathbb{P}$-a.s., then $\|X\|_\phi \le \|Y\|_\phi$; hence, we may proceed like

$$\|\rho_n - r\|_\phi \le \left\| \frac{2|\alpha_n - a|}{b} + \frac{2r|\beta_n - b|}{b} + \max_i |V_i| 1_{\{\beta_n \le b/2\}} \right\|_\phi$$

$$\le \frac{2\|\alpha_n - a\|_\phi}{b} + \frac{2|r|\|\beta_n - b\|_\phi}{b} + c|r| \|1_{\{\beta_n \le b/2\}}\|_\phi$$

$$= \frac{2\|\alpha_n - a\|_\phi}{b} + \frac{2|r|\|\beta_n - b\|_\phi}{b} + c|r|/\phi^{-1}\left(1/\mathbb{P}(\beta_n \le b/2)\right).$$

Finally, we obtain the desired result by noting that for any Young function $\chi$, $\mathbb{P}(\beta_n \le b/2) = \mathbb{P}(|(\beta_n - b)_-| \ge b/2) \le 1/\chi(b/2\|(\beta_n - b)_-\|_\chi)$. $\qquad\square$

**Theorem 10.** *Let $p \ge 1$. If $\max_i \|V_i\|_\infty \le c|r|$, then*

$$\frac{\|\rho_n - r\|_p}{|r|} \le \frac{2\|\alpha_n - a\|_p}{b} + \frac{2(1 + c)\|\beta_n - b\|_p}{b}.$$

*Proof.* Apply Lemma 9 with $\chi(x) = \phi(x) = x^p$. $\qquad\square$

**Theorem 11.** *If $|\alpha_n/\beta_n| \le 1$ $\mathbb{P}$-a.s., then*

$$|\mathbb{E}[\rho_n] - r| \le (2b^2)^{-1}\{3\mathbb{E}[(\beta_n - b)^2] + \mathbb{E}[(\alpha_n - a)^2]\}.$$

*Proof.* Using the identity

$$\frac{\alpha_n}{\beta_n} - \frac{a}{b} = \frac{\alpha_n}{\beta_n}\frac{(b - \beta_n)^2}{b^2} + \frac{(\alpha_n - a)(b - \beta_n)}{b^2} + \frac{a(b - \beta_n)}{b^2} + \frac{\alpha_n - a}{b},$$

yields

$$\mathbb{E}[\rho_n] - r = \mathbb{E}\left[\frac{\alpha_n}{\beta_n}\frac{(b - \beta_n)^2}{b^2}\right] + \frac{\mathbb{E}[(\alpha_n - a)(b - \beta_n)]}{b^2},$$

which completes the proof. $\qquad\square$

We conclude with a lemma that gives the concentration of a uniformly ergodic Markov chain. We think that this Lemma is of independent interest, and we give it under general conditions.

**Lemma 12.** *Let $(\mathbb{Z}, \mathcal{Z})$ be a state-space and $Q$ a Markov kernel on $(\mathbb{Z}, \mathcal{Z})$ which is uniformly ergodic with mixing time $t_{mix}$ and stationary distribution $\pi$. Let $(g_i)_{i=1}^n$ be a family of $\mathbb{R}^d$-valued measurable functions on $\mathbb{Z}$ such that $\|g\|_\infty = \max_{i \in \{1,\dots,n\}} \|g_i\|_\infty < \infty$ and $\pi(g_i) = 0$ for all $i \in \{1, \dots, n\}$. Then for every initial probability $\xi$ on $(\mathbb{Z}, \mathcal{Z})$, $n \in \mathbb{N}$, and $t \ge 0$,*

$$\mathbb{P}_\xi\left(\left\|\sum_{i=1}^n g_i(Z_i)\right\| \ge t\right) \le 2\exp\left(-\frac{2t^2}{u_n^2}\right), \tag{15}$$

*where $u_n = 3\|g\|_\infty \sqrt{nt_{mix}}$.*

*Proof.* The function $\varphi(x_1^{1:N}, \ldots, x_n^{1:N}) = \|\sum_{i=1}^n g_i(x_i^{1:N})\|$ on $\mathbb{Z}^n$ satisfies the bounded differences property. Applying [38, Corollary 2.10], we get, for $t \geq \mathbb{E}_\xi[\|\sum_{i=1}^n g_i(Z_i)\|]$,

$$\mathbb{P}_\xi\left(\left\|\sum_{i=1}^n g_i(Z_i)\right\| \geq t\right) \leq \exp\left\{-\frac{2(t - \mathbb{E}_\xi[\|\sum_{i=1}^n g_i(Z_i)\|])^2}{9n\|g\|_\infty^2 t_{mix}}\right\}.$$

It remains to bound $\mathbb{E}_\xi\left[\|\sum_{i=1}^n g_i(Z_i)\|\right]$ from above. For this purpose, note that

$$\mathbb{E}_\xi\left[\left\|\sum_{i=1}^n g_i(Z_i)\right\|^2\right] = \sum_{i=1}^n \mathbb{E}_\xi\left[\|g_i(Z_i)\|^2\right] + 2\sum_{k=1}^{n-1}\sum_{\ell=1}^{n-k} \mathbb{E}_\xi[g_k(Z_k)^\intercal g_{k+\ell}(Z_{k+\ell})],$$

where, using that $\pi(g_{k+\ell}) = 0$,

$$|\mathbb{E}_\xi[g_k(Z_k)^\intercal g_{k+\ell}(Z_{k+\ell})]| = \left|\int g_k(z)^\intercal \left(Q^\ell g_{k+\ell}(z) - \pi(g_{k+\ell})\right) \xi\, Q^k(\mathrm{d}z)\right| \leq \|g\|_\infty^2 (1/4)^{\lceil \ell/t_{mix}\rceil},$$

which implies that

$$\sum_{k=1}^{n-1}\sum_{\ell=1}^{n-k} |\mathbb{E}_\xi[g_k(Z_k)^\intercal g_{k+\ell}(Z_{k+\ell})]| \leq \sum_{k=1}^{n-1} \|g\|_\infty^2 (1/4)^{\lceil \ell/t_{mix}\rceil} \leq (4/3)\|g\|_\infty^2 t_{mix} n.$$

Combining the bounds above, we obtain the upper bound

$$\mathbb{E}_\xi\left[\left\|\sum_{i=1}^n g_i(Z_i)\right\|\right] \leq \left(\mathbb{E}_\xi\left[\left\|\sum_{i=1}^n g_i(Z_i)\right\|^2\right]\right)^{1/2} \leq 2\sqrt{n}\|g\|_\infty \sqrt{t_{mix}} \overset{\text{not.}}{=} v_n.$$

By plugging this result into (15), we obtain that

$$\mathbb{P}_\xi\left(\left\|\sum_{i=1}^n g_i(Z_i)\right\| \geq t\right) \leq \begin{cases} 1, & t < v_n, \\ \exp\left(-\frac{2(t-v_n)^2}{9v_n^2}\right), & t \geq v_n. \end{cases} \tag{16}$$

Now, since the right-hand side of (16) is, for every $t \geq 0$, upper bounded by $2\exp(-2t^2/(9v_n^2))$, the statement of the lemma follows. $\square$

## C Experiments

### C.1 Gaussian Mixture

**Bias MSE trade-off:** We display in Figures 4a and 4b the bias and the MSE of the BR-SNIS estimators for the same configuration as in Figure 2 but with $k_0 = \lfloor 0.625k_{max}\rfloor$. We observe 3 times less bias than the SNIS estimators but only with a 10% increase of the MSE for the $N = 129$ setting. This can be also seen in Figure 4c, where we show the ratio between BR-SNIS and SNIS for bias and MSE with $N = 129$.

**Parameters Gaussian mixture:** The $\pi$ in Section 3 is a Mixture of two Gaussians in dimension 7 with mean vectors $\boldsymbol{\mu}_1 = (1, \ldots, 1)^\intercal$ and $\boldsymbol{\mu}_2 = (-2, 0, \ldots, 0)^\intercal$ and covariance matrices $\boldsymbol{\Sigma}_1 = d^{-1}\mathbf{I}$ and $\boldsymbol{\Sigma}_2 = d^{-1}\mathbf{I}$, where $p = 1/3$ and $\mathbf{I}$ is the identity matrix In this setting, the quantities $\kappa[\pi, \lambda]$ and $\omega$ can be estimated by Monte Carlo and Gradient ascent respectively. Their values are approximately $7 \cdot 10^2$ and $1 \cdot 10^4$, respectively.

The sets $A$ and $B$ used to define the function $f$ are the following:

$$A := [-2, 6] \times [-1, 1]^6, \quad B := [0.75, 1.25] \times [1, 2] \times [-0.1, 0.1]^5.$$

We used this example to illustrate numerically the bounds in Theorems 3 and 4, where each expectation was calculated by Monte Carlo using $2 \cdot 10^4$ samples. We displayed in each figure the equivalent SNIS estimation in a green dashed line. For all the bias related bounds(Theorem 3(i) in Figure 5a, Theorem 4(i) in Figure 5c), we fixed a total budget of $M = 6 \cdot 10^3$. For Figure 5a we added a fit of the type $y = \exp(ak + b)$ to illustrate the exponential decay w.r.t. $k$.

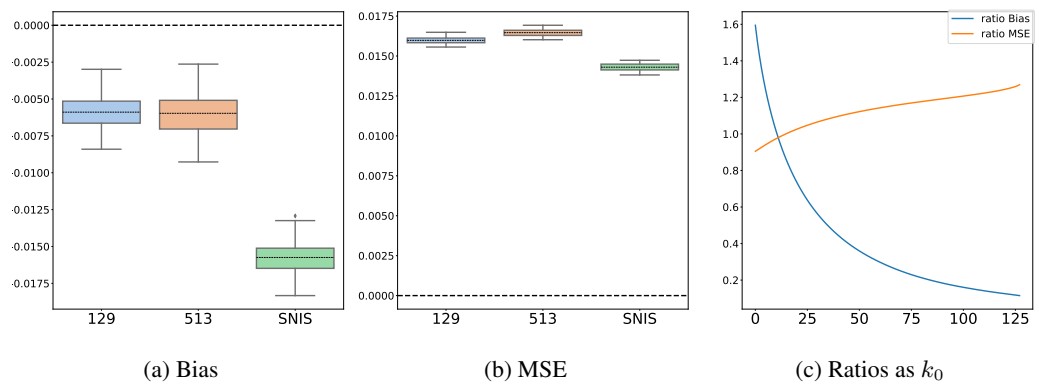

(a) Bias        (b) MSE        (c) Ratios as $k_0$

Figure 4: Comparison between SNIS and BR-SNIS for the same budget. In each boxplot the dotted line represents the **mean** value of the samples. In Figure 4c we display the ratio between BR-SNIS and SNIS for bias and MSE with $N = 129$.

We then increased the budget to $M = 8 \cdot 10^4$ for the MSE and covariance bounds, in order to fully observe the stabilisation of the MSE in Figure 5b for all the minibatch sizes $N$. For the true value of $\pi(f)$ needed for calculating the MSE, we use an estimation obtained by Monte Carlo (sampling directly from $\pi$) with $4 \cdot 10^7$ samples. In Figure 5d we added dashed lines with the theoretical value of the $\mathsf{MSE}^{is}_{\upsilon M}$ with the same color as $\upsilon$.

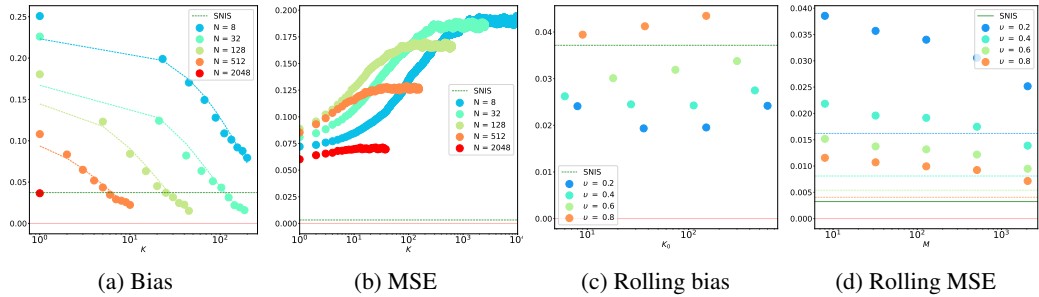

(a) Bias      (b) MSE      (c) Rolling bias      (d) Rolling MSE

Figure 5: Visualization of the theoretical bounds from Theorems 3 and 4.

**Comparison with zero bias SNIS methods:** There exists estimators based on SNIS that have no bias, such as the estimator proposed in [33] and refered to as Unbiased-PIMH . One of the main differences between such estimator and BR-SNIS is that BR-SNIS works under a pre-established budget of samples, whereas in Unbiased-PIMH the number of samples used to produce an estimate varies due to the accept-reject procedure. Even though the two estimators have different goals, it can be of interest to compare both of them in the case where there is a restriction in the total number of samples available.

We proceed to a fixed-budget ($M$) comparison between BR-SNIS and the "Rao Blackwellized" version of the algorithm proposed at [33] in the Gaussian Mixture example. In order to do so, it's necessary to impose the fixed-budget constraint to the Unbiased-PIMH estimator. A single iteration of the estimator from Unbiased-PIMH with batch-size $N$ needs $rN$ samples where $r \in \mathbb{N}$ is a random number satisfying $r \geq 2$. Therefore, there are two ways of applying the constraint to Unbiased-PIMH :

- **Soft**: For a given $N$, generate estimations using Unbiased-PIMH until the number of samples is larger than $M$ and **keep** the last estimation. Therefore, all the estimators from Unbiased-PIMH will have used **at least** $M$ samples. All the estimations generated are averaged to generate a single estimate.

- **Hard**: For a given $N$, generate estimations using Unbiased-PIMH until the number of total samples used is larger than $M$ and **discard** the last estimation. Therefore, all the

estimators from Unbiased-PIMH will have used **at most** $M$ samples. **If no estimations were produced under the budget cap (first iteration used more than $M$ samples), then we consider it a miss**. All the estimations generated are averaged to create a single estimate.

The code used to run the experiments is available at [2]. For both cases, the following values of $M$ are used in the comparison: $2^{16}, 2^{12}, 2^9$. For each estimator, a total of $1024$ Monte Carlo replications are used to estimate the mean and the standard deviation of the estimator. Note that in the **Hard** framework, **it can happen that less than 1024 replications are used for the Unbiased-PIMH estimator**. The number of failed estimations is reported in the tables for the framework **Hard** for each configuration.

For each configuration of the BR-SNIS estimator (defined by $N$, $k_{max}$), we have used $90\%$ burn-in period ($k_0 = \lfloor 0.9 k_{max} \rfloor$) and $k_{max}$ rounds of bootstrap ($k_{max}$ permutations of the input samples).

The following values were calculated:

- **Bias**: The mean of the estimations minus $\mathrm{ref}$ over $1024$ replications
- **Std**: The standard deviation of the estimations over $1024$ replications.
- **Fails**: The number of replications that failed to produce a single estimation for a given budget $M$. This is only applicable for the Unbiased-PIMH estimator and in the **Hard** framework.
- **average M**: The average (over the $1024$ replications) total cost of the estimator. For BR-SNIS and SNIS this is always $M$. For Unbiased-PIMH in the **Soft** framework it is larger than $M$. In the **Hard** framework it is smaller than $M$.

---

**Algorithm 1:** Unbiased-PIMH

---
**Data:** $N \geq 0$

1   $e_1, \mathrm{lwav}_1 \leftarrow \mathrm{SNIS}(N)$ ;      /* SNIS also returning the average log weights */
2   $e_2, \mathrm{lwav}_2 \leftarrow \mathrm{SNIS}(N)$;
3   **if** $\mathrm{lwav}_1 < \mathrm{lwav}_2$ **then**
4      |   $\mathrm{swap}(e_1, \mathrm{lwav}_1; e_2, \mathrm{lwav}_2)$
5   **end**
6   $u = \log \mathrm{rand}()$ ;
7   **if** $u < \mathrm{lwav}_1$ *and* $u < \mathrm{lwav}_2$ **then**
8      |   $\tau = 1$;
9   **end**
10   $t \leftarrow 1$;
11   $\tau = \infty$;
12   **while** $\tau = \infty$ **do**
13      |   $e_1 = e_1 + (e_1 - e_2)$ ;
14      |   $e_p, \mathrm{lwav}_p = \mathrm{SNIS}(N)$;
15      |   $t = t + 1$;
16      |   $u = \log \mathrm{rand}()$; **if** $u < \mathrm{lwav}_p - \mathrm{lwav}_1$ **then**
17      |      |   $e_1, \mathrm{lwav}_1 = e_p, \mathrm{lwav}_p$;
18      |   **end**
19      |   **if** $u < \mathrm{lwav}_p - \mathrm{lwav}_1$ **then**
20      |      |   $e_2, \mathrm{lwav}_1 = e_p, \mathrm{lwav}_p$;
21      |   **end**
22      |   **if** $u < \mathrm{lwav}_1$ *and* $u < \mathrm{lwav}_2$ **then**
23      |      |   $\tau = t$;
24      |   **end**
25   **end**

---

We have compared both estimators in two different frameworks (**Hard** and **Soft**) with three different budgets $M = 2^{16}$ (tables 3 and 6), $M = 2^{12}$ (tables 4 and 7) and $M = 2^9$ (tables 5 and 8). We observed that in general the BR-SNIS estimator has smaller standard deviation, with the difference of

---

[2]https://github.com/gabrielvc/br_snis/blob/master/notebooks/Comparison_Unbiased-PIMH.ipynb

| N | k | algorithm | Bias | std | average M |
|---|---|---|---|---|---|
| 65536 | | SNIS | -0.0029 | 0.0605 | 65536.0 |
| 65 | 1024 | BR-SNIS | -0.0010 | 0.0658 | 65536.0 |
| 129 | 512 | BR-SNIS | -0.0006 | 0.0689 | 65536.0 |
| 257 | 256 | BR-SNIS | 0.0003 | 0.0678 | 65536.0 |
| 513 | 128 | BR-SNIS | 0.0019 | 0.0670 | 65536.0 |
| 16384 | | Unbiased-PIMH | 0.0065 | 0.1005 | 71904.0 |
| 8192 | | Unbiased-PIMH | 0.0058 | 0.1066 | 71040.0 |
| 4096 | | Unbiased-PIMH | 0.0082 | 0.1139 | 69316.0 |
| 2048 | | Unbiased-PIMH | 0.0053 | 0.1174 | 67764.0 |

Table 3: $M = 2^{16}$ in the **Soft** framework.

| N | k | algorithm | Bias | std | average M |
|---|---|---|---|---|---|
| 4096 | | SNIS | -0.0365 | 0.1946 | 4096.0 |
| 65 | 64 | BR-SNIS | -0.0314 | 0.2211 | 4096.0 |
| 129 | 32 | BR-SNIS | -0.0358 | 0.2214 | 4096.0 |
| 257 | 16 | BR-SNIS | -0.0281 | 0.2282 | 4096.0 |
| 513 | 8 | BR-SNIS | -0.0296 | 0.2351 | 4096.0 |
| 1024 | | Unbiased-PIMH | 0.0587 | 0.6073 | 5388.0 |
| 512 | | Unbiased-PIMH | 0.0678 | 0.8086 | 5027.5 |
| 256 | | Unbiased-PIMH | 0.1258 | 1.1492 | 4730.0 |
| 128 | | Unbiased-PIMH | 0.2364 | 1.9521 | 4629.6 |

Table 4: $M = 2^{12}$ in the **Soft** framework.

| N | k | algorithm | Bias | std | average M |
|---|---|---|---|---|---|
| 512 | | SNIS | -0.1458 | 0.2420 | 512.0 |
| 65 | 8 | BR-SNIS | -0.1537 | 0.2468 | 512.0 |
| 129 | 4 | BR-SNIS | -0.1543 | 0.2444 | 512.0 |
| 257 | 2 | BR-SNIS | -0.1426 | 0.2600 | 512.0 |
| 128 | | Unbiased-PIMH | -0.0048 | 1.3924 | 841.5 |
| 64 | | Unbiased-PIMH | 0.1997 | 2.5677 | 796.4 |
| 32 | | Unbiased-PIMH | 0.2365 | 4.1642 | 708.1 |
| 16 | | Unbiased-PIMH | 0.3670 | 5.1533 | 685.3 |

Table 5: $M = 2^9$ in the **Soft** framework.

| N | k | algorithm | Bias | std | average M | Fails |
|---|---|---|---|---|---|---|
| 65536 | | SNIS | -0.0029 | 0.0605 | 65536.0 | |
| 65 | 1024 | BR-SNIS | -0.0006 | 0.0650 | 65536.0 | |
| 129 | 512 | BR-SNIS | -0.0023 | 0.0645 | 65536.0 | |
| 257 | 256 | BR-SNIS | -0.0024 | 0.0657 | 65536.0 | |
| 513 | 128 | BR-SNIS | 0.0000 | 0.0693 | 65536.0 | |
| 16384 | | Unbiased-PIMH | -0.0028 | 0.0885 | 57520.0 | 7 |
| 8192 | | Unbiased-PIMH | -0.0008 | 0.1029 | 59264.0 | 0 |
| 4096 | | Unbiased-PIMH | -0.0014 | 0.1026 | 61956.0 | 0 |
| 2048 | | Unbiased-PIMH | 0.0008 | 0.1106 | 63244.0 | 0 |

Table 6: $M = 2^{16}$ in the **Hard** framework.

| N | k | algorithm | Bias | std | average M | Fails |
|---|---|---|---|---|---|---|
| 4096 | | SNIS | -0.0365 | 0.1946 | 4096.0 | |
| 65 | 64 | BR-SNIS | -0.0252 | 0.2270 | 4096.0 | |
| 129 | 32 | BR-SNIS | -0.0296 | 0.2221 | 4096.0 | |
| 257 | 16 | BR-SNIS | -0.0338 | 0.2218 | 4096.0 | |
| 513 | 8 | BR-SNIS | -0.0486 | 0.2243 | 4096.0 | |
| 1024 | | Unbiased-PIMH | -0.0901 | 0.2353 | 2922.0 | 103 |
| 512 | | Unbiased-PIMH | -0.0833 | 0.3368 | 3343.0 | 24 |
| 256 | | Unbiased-PIMH | -0.0547 | 0.4815 | 3554.8 | 9 |
| 128 | | Unbiased-PIMH | -0.0634 | 0.4433 | 3683.1 | 4 |

Table 7: $M = 2^{12}$ in the **Hard** framework.

| N | k | algorithm | Bias | std | average M | Fails |
|---|---|---|---|---|---|---|
| 512 | | SNIS | -0.1458 | 0.2420 | 512.0 | |
| 65 | 8 | BR-SNIS | -0.1376 | 0.2636 | 512.0 | |
| 129 | 4 | BR-SNIS | -0.1456 | 0.2565 | 512.0 | |
| 257 | 2 | BR-SNIS | -0.1358 | 0.2585 | 512.0 | |
| 128 | | Unbiased-PIMH | -0.1962 | 0.2200 | 306.9 | 210 |
| 64 | | Unbiased-PIMH | -0.1947 | 0.3200 | 367.8 | 73 |
| 32 | | Unbiased-PIMH | -0.1999 | 0.4001 | 398.0 | 36 |
| 16 | | Unbiased-PIMH | -0.2057 | 0.7366 | 423.2 | 16 |

Table 8: $M = 2^9$ in the **Hard** framework.

standard deviation being important for the smaller budgets (3 times less for $M = 2^{12}$ and 10 times less for $M = 2^9$ in the **Soft** framework).

For the **Hard** framework, we can see that the empirical bias of BR-SNIS is always at most equal to the empirical bias of Unbiased-PIMH . For the **Soft** framework, we observed that for $M = 2^{16}$ that both methods have similar performance, with BR-SNIS having negligible bias in this setting. For $M = 2^{12}$ and $M = 2^9$, BR-SNIS has in general a smaller empirical biais and the standard deviation of Unbiased-PIMH is considerably higher.

### C.2 Bayesian Logistic regression

The importance distribution used in the Bayesian logistic regression example is given by the mean-field variational distribution [6]. More precisely, given the target $\pi$ given in Section 3, the proposal $\lambda$ is a Gaussian distribution with mean $\mu$ and diagonal covariance $\mathrm{diag}(\sigma)$, where $\mu, \sigma$ are learnt by maximization of the Evidence Lower Bound (ELBO):

$$\mathcal{L}(\mu, \sigma) = \int \log(\pi(\theta)/\lambda(\theta))\lambda(\theta)\mathrm{d}\theta.$$

In both Figures 3 and 6, the optimal $k$ for a given budget $M$ was chosen by grid search over all the factors of $M$. The final settings are shown in Table 9.

### C.3 Importance Weighted Auto-Encoders

We trained each network for a total of 100 epochs, using 512 batch samples for the gradient calculations, with learning rate equals $10^{-4}$. For IWAE and BR-IWAE , 64 samples were used for estimating the gradient. For BR-IWAE, we used $k = 8$. The architecture used is described in table 10 where by conv layer we mean a convolutional layer followed by batch norm and the ReLU activation function. The train ELBO for each latent dimension is shown in Figure 7. For the log likelihood comparison in Table 2, we use SNIS with the variational posterior as importance distribution and a total of $2 \cdot 10^3$ samples for a subset of 3232 samples from the validation set. Therefore, the estimation of the log likelihood is:

$$\hat{\mathcal{L}} = T^{-1} \sum_{j=1}^{T} \sum_{i=1}^{M} \omega_{\theta,\phi,x_j} \log p_\theta(x_j \mid z_i^j)$$

with $\omega_{\theta,\phi,x}(z) = p_\theta(x)/q_\phi(z \mid x)$ where $z_i^j$ is sampled from $q_\phi(\cdot \mid x_j)$.

| Dataset | component | M | $k_{max}$ | N |
|---|---|---|---|---|
| breast | 8 | 256 | 4 | 65 |
| breast | 8 | 512 | 8 | 65 |
| breast | 8 | 1024 | 16 | 65 |
| breast | 8 | 2048 | 16 | 129 |
| breast | 8 | 4096 | 64 | 65 |
| breast | 11 | 256 | 4 | 65 |
| breast | 11 | 512 | 8 | 65 |
| breast | 11 | 1024 | 16 | 65 |
| breast | 11 | 2048 | 32 | 65 |
| breast | 11 | 4096 | 64 | 65 |
| breast | 14 | 256 | 4 | 65 |
| breast | 14 | 512 | 8 | 65 |
| breast | 14 | 1024 | 16 | 65 |
| breast | 14 | 2048 | 32 | 65 |
| breast | 14 | 4096 | 64 | 65 |
| heart | 5 | 32 | 4 | 9 |
| heart | 5 | 64 | 8 | 9 |
| heart | 5 | 128 | 8 | 17 |
| heart | 5 | 256 | 32 | 9 |
| heart | 5 | 512 | 4 | 129 |
| heart | 8 | 32 | 4 | 9 |
| heart | 8 | 64 | 8 | 9 |
| heart | 8 | 128 | 8 | 17 |
| heart | 8 | 256 | 16 | 17 |
| heart | 8 | 512 | 32 | 17 |
| heart | 12 | 32 | 4 | 9 |
| heart | 12 | 64 | 8 | 9 |
| heart | 12 | 128 | 16 | 9 |
| heart | 12 | 256 | 4 | 65 |
| heart | 12 | 512 | 32 | 17 |
| covertype | 6 | 512 | 4 | 129 |
| covertype | 6 | 1024 | 8 | 129 |
| covertype | 6 | 2048 | 16 | 129 |
| covertype | 6 | 4096 | 2 | 2049 |
| covertype | 6 | 8192 | 4 | 2049 |
| covertype | 17 | 512 | 2 | 257 |
| covertype | 17 | 1024 | 2 | 513 |
| covertype | 17 | 2048 | 2 | 1025 |
| covertype | 17 | 4096 | 2 | 2049 |
| covertype | 17 | 8192 | 4 | 2049 |
| covertype | 23 | 512 | 2 | 257 |
| covertype | 23 | 1024 | 2 | 513 |
| covertype | 23 | 2048 | 4 | 513 |
| covertype | 23 | 4096 | 16 | 257 |
| covertype | 23 | 8192 | 32 | 257 |

Table 9: Optimal configurations for Figures 3 and 6

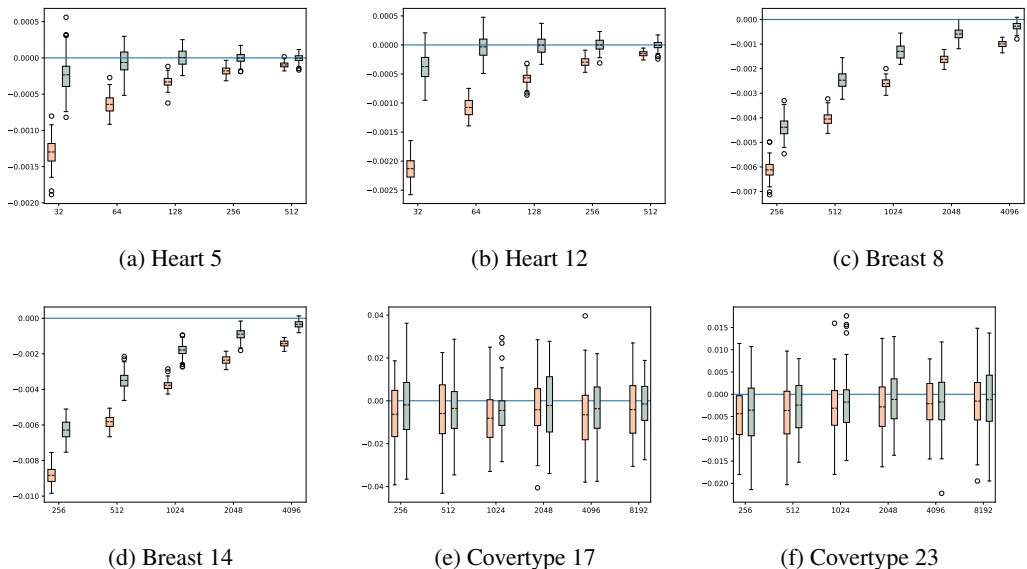

(a) Heart 5           (b) Heart 12           (c) Breast 8

(d) Breast 14           (e) Covertype 17           (f) Covertype 23

Figure 6: Visualisation of the distribution of the bias for the Heart Failure and Breast cancer dataset for other components of $\theta$

| Name | kernel size | stride | padding | out channels |
|---|---|---|---|---|
| Encoder conv 1 | 3 | 1 | 1 | 8 |
| Encoder conv 2 | 3 | 1 | 1 | 16 |
| Encoder conv 3 | 3 | 1 | 1 | 32 |
| Encoder MaxPool2d 1 | 2 | 2 | 0 | |
| Encoder conv 4 | 3 | 1 | 1 | 64 |
| Encoder conv 5 | 3 | 1 | 1 | 32 |
| Encoder MaxPool2d 2 | 2 | 2 | 0 | |
| Encoder Linear + ReLU | | | | 2048 |
| Encoder Linear | | | | $2 * d$ |
| | | | | |
| Decoder Linear | | | | $32 * 7 * 7$ |
| Decoder conv transpose 1 | 2 | 1 | 0 | 64 |
| Decoder conv transpose 2 | 2 | 1 | 1 | 128 |
| Decoder conv transpose 3 | 3 | 2 | 1 (output padding = 1) | 64 |
| Decoder conv transpose 4 | 3 | 2 | 1 (output padding = 1) | 32 |
| Decoder conv transpose 5 | 2 | 1 | 0 | 16 |
| Decoder final convolutional layer | 2 | 1 | 0 | 1 |
| Sigmoid activation | | | | |

Table 10: Convolutional neural network architecture.

## C.4   Resources

All the simulations were done using a server with the following configuration:

- GPUs: two Tesla V100-PCIE (32Gb RAM)
- CPU: 71 Intel(R) Xeon(R) Gold 6154 CPU @ 3.00GHz
- RAM: 377Gb

locally hosted. We estimate the total number of computing hours for the results presented in this paper to be inferior to 200 hours of GPU usage (All the calculations were done in the GPU).

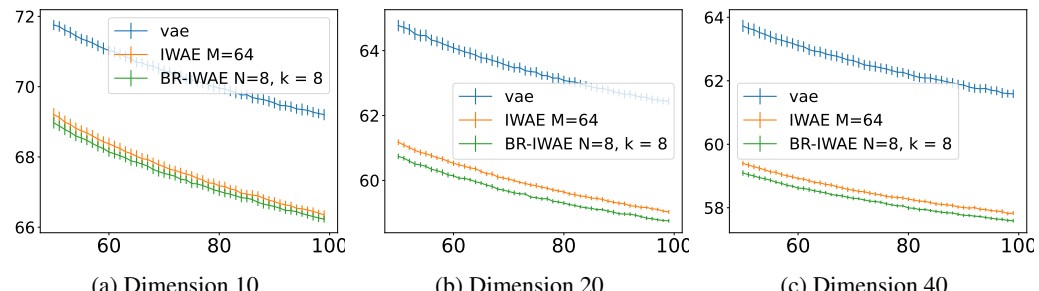

Figure 7: Per epoch training loss (ELBO) for the last 40 epochs. Confidence intervals are calculated as $1.96\sigma/\sqrt{n}$ over 10 ($n = 10$) different seeds.