# OpenReview forum: "BR-SNIS: Bias Reduced Self-Normalized Importance Sampling"
_NeurIPS.cc/2022/Conference — NeurIPS 2022 Accept_

### Official Review · Reviewer_hWyK · 2022-07-11

**Rating:** 5
**Confidence:** 3
**Soundness:** 2 fair
**Presentation:** 2 fair
**Contribution:** 2 fair

**Summary:**

The iterated sampling importance resampling (i-SIR) is an iterative IS estimator that at each iteration a pool of candidate particles consists of one resample particle from the previous iteration and particles generated from an importance function. The paper suggests to average estimators over k-iterations instead of taking an estimator at the last iteration to reduce the bias of estimator and make better use of the available computational resources. Then theoretical properties were studied and simulation studies are given in comparison to a naïve self-normalized IS.

**Questions:**

Please see Strengths and Weakness section.

**Limitations:**

yes

**Strengths And Weaknesses:**

Most of literature tend to focus on an importance function to make a better estimator. In that sense the study on i-SNR is interesting. Without too much difficulty, it is easy to see that averaging independent multiple IS estimates will reduce the bias compared to an estimate from a single iteration. Although the i-SNR is slight different to independent ISs, the novelty in methodology seem to be not big.

The evidence supporting the efficient computation does not look sufficient and I feel the method is needed to be fully investigated both theoretically and numerically in general.

a) The expectation error bound in Theorem 4 looks increasing with k (number of iterations). The burn-in k0 is likely to depend on various things like the deviation of an importance function to the target distribution and a number of samples per iteration. It sounds strange.
b) How do you find the burn-in k0?
c) In the experiment, k0=k-1. This mean that an estimate from the last iteration is taken and this against the claim that the new approach makes better use of computation by recycling all candidates.
d) Given the computation budget, there will be some trade-off between a number of samples N and k (iterations). Some investigation will be good.
e) Numerical comparison with other similar estimator (iterative IS estimator) will make the comparison more fair and useful. For example, Population Monte Carlo method.
Cappe, O., Guillin, A., Marin, J. M> and Robert, C. P. (2004) Population Monte Carlo, Journal of Computational and Graphical Statistics. 13(4) 907-929.
f) For simulation study with logistic regressions, what are the computing  budget M and k?

---

> ### Author Response · Authors · 2022-07-30
> **A wrapper of SNIS and not yet another MCMC algorithm as well as Question (e)**
>
> What we read shows a lack of understanding of what we are proposing. Our presentation is probably not clear enough and we will rework the introduction. **We are not developing an MCMC algorithm, but a wrapper** to reduce SNIS bias (while tightly controlling the increase in variance), all at very low algorithmic cost. **We do not propose a new variant of i- SIR algorithm**.
>
> To compute an SNIS estimator, we need to simulate $M$ samples under the proposal law and compute $M$ importance weights. The bias and variance of the SNIS estimator are inversely proportional to $M$ (with explicit constants). Our estimator computes the same quantities but uses them differently, with the goal of reducing bias.
> We then add an "extra-randomization" as follows.
> - We divide the set of samples into $k$ blocks of size $N-1$ with $M=k(N-1)+1$.
> - We select a sample and then use the i-SIR algorithm with the previously calculated importance weights (with no need of simulating new points or recalculating importance ratios).
> - We eliminate the first $k_0$ blocks (burn-in) and then compute an estimator by recycling all samples over the remaining $(k-k_0)$ blocks.
> - We repeat this process $\ell$ times after randomly permuting the data and finally combine these estimators [bootstrap procedure].
> The cost of the procedure is very limited (as most of the computational cost in the examples that we are dealing with comes from the simulation under the importance law and the calculation of the importance weights).
>
> Question (e): The goal of this article is not to propose a new method for importance sampling. There is, of course, an extensive literature on adaptive importance sampling methods, but that is absolutely not the goal of the present paper.

---

> > ### Comment · Reviewer_hWyK · 2022-08-05
> > **Thank you.**
> >
> > Thank you for the detailed answers. I have changed my score from 4 to 5.

---

> ### Author Response · Authors · 2022-07-30
> **Questions (a), (b), (c), and (d)**
>
> (a): The fact that the variance is inversely proportional to $(k-k_0) N= M$ is the "expected" result (which, incidentally, is not trivial). If we eliminate $k_0$ blocks of size $N-1$ for the burn-in, we are left with $(k-k_0)(N-1)$ samples on which to compute the estimator. Note: $k$ here is the total number of blocks of size $N-1$ in the sample of size $M$. Thus, **there is nothing strange here**. The variance is then reduced by the bootstrap procedure (if we iterate the procedure).
>
> (b) and (c): In the first set of simulations, we have set $k_0= (k-1)$ because it is simple (we only have to choose the block size $N$) and maximizes the bias-reduction effect. **All the samples are recycled using the bootstrap procedure**:  we randomly swap the data, and repeat the procedure. This is the bootstrap that ensures we use all the data. Other choices are possible, as illustrated in our numerical experiments.  We have not developed automatic procedures for choosing $k_0$, but it is clear that we can take inspiration from the explicit bounds here.
>
> As illustration, we have performed an additional simulation for the elementary mixture model.
> | Ratio Bias  |  Ratio  MSE | Burn-in (k_0 / k) |
> | :----:  |  :----:  | :----: |
> |0.644          |	1.048    |	0.200|
> |0.355 	    | 1.125    |	0.400|
> |0.229	    |1.172    |	0.600|
> |     0.159     |    1.211    |	0.800|
>
> "Ratio Bias" is the bias of BR-SNIS divided by the bias of SNIS. "Ratio MSE" is the MSE of BR-SNIS divided by the MSE of SNIS. The burn-in percentage is the percentage ($\upsilon$ in our work) of proposed samples removed for each bootstrap sample. Here, the minibatch size ($N$ in the paper) is set to 129, while all other sizes remain as specified in the paper. The number of bootstrap replicates is 128.
> Clearly, larger $k_0$ yields larger bias reduction. The increase of variance is moderate due to bootstrap.
>
> We updated the figures in the manuscript to focus more clearly on bias reduction. We have also added a figure illustrating the tradeoff between reducing the bias and increasing MSE.

---

> ### Author Response · Authors · 2022-07-30
> **Question (f)**
>
> In the appendix of the revised version of our paper, we have added a table with the budget $M$ and number $k$ of iterations for all the data presented in the logistic regression study.
> We show here an extract of the table for the estimation of the 8-th coordinate of the posterior mean vector for the heart dataset. We also add the following two quantities:
>
> * "Ratio bias": bias of BR-SNIS divided by the bias of SNIS,
> * "Ratio MSE": MSE of BR-SNIS divided by the MSE of SNIS.
>
> |Dataset | M | k | N | Ratio bias| Ratio MSE |
> | :---: | :---: | :---: | :---: | :---: | :---: |
> |heart|32|4|9| 0.18 | 0.05 |
> |heart|64|8|9|0.03 | 0.05|
> |heart|128|8|17|0.005 | 0.08|
> |heart|256|16|17|0.025 | 0.14|
> |heart|512|32|17|0.07|0.23|

---

### Official Review · Reviewer_1FgZ · 2022-07-11

**Rating:** 5
**Confidence:** 4
**Soundness:** 3 good
**Presentation:** 3 good
**Contribution:** 3 good

**Summary:**

The paper proposes an MCMC-based extension of self-normalized importance sampling which can be seen as an iterated SIR algorithm or as a single-time-step analogue of particle MCMC. The interest in the current setting is to use a "waste recycling" variant of the iSIR algorithm in the SNIS context in order to provide a variance reduction.

A thorough theoretical analysis is presented which it is claimed support the idea that this approach may lead to "drastic bias reduction without impairing the variance".

**Questions:**

1. I've been unable to reconcile the claim in the abstract that the proposed method reduces bias without increasing variance with the numerical results shown in Figure 2. MSE = bias^2 + variance and the proposed method seemingly increases the MSE relative to the basic SNIS method which would be consistent with it increasing the variance by somewhat more than the reduction in (squared) bias. Is that the case? If so the abstract is a little misleading, if not some explanation seems necessary.

2. Does the reduction in bias lead to meaningful improvements in practical problems that the authors have seen?

3. Is the work of Frenkel or Andrieu et al. on "waste recycling" relevant prior art from a methodological perspective or is there a reason that it is not?

**Limitations:**

The authors indicated that their work is of a theoretical/methodological nature and hence did not consider societal impact. I think that is reasonable.



**Strengths And Weaknesses:**

I like the attempt to improve on a basic algorithm using a simple "plug-in" strategy which should give the proposed method wide applicability. The main strengths of the paper are probably casting this problem in a framework which should allow practitioners to readily adapt existing SNIS implementations to make use of the proposed approach and the rigorous theoretical treatment of the method which provides some confidence in the performance of the approach.

In terms of weaknesses, there seem to me to be two:

1. The claimed novelty seems to me to be overly broad.
2. It is not immediately clear that one obtains the "drastic" improvements that are reapeatedly claimed in a way which leads to any practical benefit.

In terms of novelty, I was surprised to see it suggested that the idea of recycling "rejected" samples in this setting was somehow an innovation as it's an idea I've seen discussed regularly at conferences. Indeed, looking at Section 4.6 of [Andrieu, Doucet and HOlenstein. Particle Markov chain monte Carlo methods. Journal of the Royal Statistical Society Series B, 72(3):269--342 2010] or [Frenkel, D. (2006) Waste-recycling Monte Carlo. Lect. Notes Phys., 703, 127–138.] demonstrate that this is something which has been reasonably well known and in the literature for some time -- and with some formal justification.

In terms of improvement, there are two issues which the manuscript doesn't presently do justice to in my mind. These are:
1. Is there really a reduction in variance at no cost; seemingly, the variance increases by more than the squared bias falls in all of the empirical results and the theoretical results seem to provide a bound on the amount by which the MSE can increase rather than establishing as claimed in the abstract that there is no increase.
2. Does the reduction in bias make meaningful practical differences? The last numerical example seems to come closest to showing that it leads to an improvement in a widely used figure of merit but it stops short of showing improved, say, predictive performance. Although it might seem obvious that reducing the bias would improve performance it seems to me not to be so largely because bias in SNIS vanishes at the same rate of variance and hence typically makes negligible contribution to MSE for even modest numbers of samples and the MSE estimates shown for the proposed algorithm seem marginally worse than those for SNIS.
These are both important questions in assessing how relevant the work is to the NeurIPS community.

---

> ### Author Response · Authors · 2022-07-30
> **Bias and variance—general comments**
>
> First, thank you for your very constructive review, which will allow us to improve the presentation of the paper. In particular, in the new version of the paper, we have now updated the experimental section.
>
> - We do not claim that the proposed method reduces the variance. More precisely, we have a bound on the variance, whose main term remains inversely proportional to the number of simulated samples (with the same constant as for the SNIS estimator). Moreover, the procedure for reducing the bias leads to a controlled increase in variance, as we report in our experiments and in the table of results below (for logistic regression).
> - Reduction of bias **makes a difference**! Of course, reducing the bias is relevant primarily from a frequentist point of view. Indeed, if we only draw a single realization once, only the distribution of the variable (and thus typically the bias and the variance) matters. The bias of an estimator becomes important in situations where the estimator is used multiple times, as in stochastic approximation procedures; see
> [Doucet, A., & Tadic, V. B. (2017). Asymptotic bias of stochastic gradient search. Annals of Applied Probability, 27(6).]
> or
> [Karimi, B., Miasojedow, B., Moulines, E., & Wai, H. T. (2019). Non-asymptotic analysis of biased stochastic approximation scheme. In Conference on Learning Theory (pp. 1944-1974). PMLR.]
> In stochastic approximation, the bias and variance of the mean-field estimates do not play the same role in the bounds. For this reason, it is interesting to reduce bias (provided the variance does not explode) for a given computational budget, and there are also reasons why reducing bias (or obtaining an unbiased estimator) has given rise to much work related to simulation algorithms over the past 30 years; see for example [Glynn, P. W., & Rhee, C. H. (2014). Exact estimation for Markov chain equilibrium expectations. Journal of Applied Probability, 51(A), 377-389.] and [Jacob, P. E., O’Leary, J., & Atchadé, Y. F. (2020). Unbiased Markov chain Monte Carlo methods with couplings. Journal of the Royal Statistical Society: Series B (Statistical Methodology), 82(3), 543-600.]
> We will add a discussion to the revised version of our paper with the aim of convincing the reader that reducing bias is indeed a reasonable goal when estimators are used repeatedly, but that this does not make much sense when the estimator is computed only once.

---

> > ### Comment · Reviewer_1FgZ · 2022-08-07
> > **Response**
> >
> > Thank you for taking the timing to respond in detail. The point about bias contributing differently to variance in contexts other than mean squared error is certainly well taken, I would have liked to see this point made more clearly in the manuscript and am please to hear that you intend to do so.

---

> > > ### Author Response · Authors · 2022-08-09
> > > **Thanks for the suggestion**
> > >
> > > Thanks for your feedback. Following your suggestion, we have now added—a short version of—this discussion to Section 2.3.

---

> ### Author Response · Authors · 2022-07-30
> **Bias--variance trade-off illustrated in a specific case**
>
> First of all we would like to thank you for raising these important points, which will make the paper's message clearer.
> We will address the following point in this comment:
> * "Is there really a reduction in variance at no cost; seemingly, the variance increases by more than the squared bias falls in all of the empirical results and the theoretical results seem to provide a bound on the amount by which the MSE can increase rather than establishing as claimed in the abstract that there is no increase."
>
> For the mixture-of-Gaussians example (first example) presented in our paper, the bias$^2$ (bias$^2$ of the SNIS  estimator is of order $10^{-4}$ in this case) is much smaller than the MSE ($10^{-2}$); this is not surprising, since the bias as well as the variance scale inversely proportionally to the number of proposed samples.  Nevertheless, we claim that we can achieve a significant reduction of the bias with only a moderate increase of the MSE (or, equivalently, the variance). To make this claim more explicitly, we provide the following table:
>
> | Ratio Bias  |  Ratio  MSE | Burn-in (k_0 / k) |
> | :----:  |  :----:  | :----: |
> |0.644          |	1.048    |	0.200|
> |0.355 	    | 1.125    |	0.400|
> |0.229	    |1.172    |	0.600|
> |     0.159     |    1.211    |	0.800|
>
> Ratio bias is the bias of BR-SNIS divided by the bias of SNIS. Ratio MSE is the MSE of BR-SNIS divided by the MSE of SNIS. The burn-in percentage is the percentage ($\upsilon$ in our work) of proposed samples removed for each bootstrap sample. Here, the minibatch size ($N$ in the paper) is set to 129, while all other sizes remain as specified in the paper. The number of bootstrap replicates is 128.
>
> We updated the figures in the manuscript to focus more clearly on bias reduction. We have also added a figure illustrating the tradeoff between reducing the bias and increasing the MSE.
>
> Maximizing the bias reduction prompts us to choose $k_0$ large. In the previous example, we obtain a reduction of bias by a factor of 10 by taking $k_0= (4/5) k$. This choice leads to an increase in variance by a factor 5 for a single replication. The bootstrap procedure reduces this increase to a factor of 1.2; thus, the bootstrap procedure allows us to reduce the bias fairly aggressively while avoiding a large increase in the variance.
>
> Figure 1(a) now reports the evolution of the MSE as a function of $k$ and the number of bootstrap rounds (1, 21, and 201).

---

### Official Review · Reviewer_nbwq · 2022-07-12

**Rating:** 6
**Confidence:** 3
**Soundness:** 3 good
**Presentation:** 2 fair
**Contribution:** 2 fair

**Summary:**

The authors describe and analyze an iterated importance sampling resampling scheme  and a possible bias-reduction technique.

**Questions:**

The paper contains very interesting material, in my opinion. I believe that the degree of novelty is mainly in the bias reduction idea. However, I have also some concerns. See below.

- However, the relationships and connections  with MCMC methods with multiple candidates schemes such as the Ensemble MCMC algorithms in

R. Neal, MCMC using ensembles of states for problems with fast and slow variables such as Gaussian process regression, arXiv:1101.0387, 2011.

B. Calderhead, A general construction for parallelizing Metropolis-Hastings algorithms, Proceedings of the National Academy of Sciences of the United States of America (PNAS) 111 (49) (2014) 17408–17413.

E. Bernton, S. Yang, Y. Chen, N. Shephard, J. S. Liu, Locally weighted Markov Chain Monte Carlo, arXiv:1506.08852 (2015) 1–14.

should be discussed. An interesting summary and review of these techniques is given in Section 4.4 of

L. Martino, "A Review of Multiple Try MCMC algorithms for Signal Processing", Digital Signal Processing, Volume 75, Pages: 134-152, 2018.

These techniques are very similar, in my opinion, to the method you are describing, hence this is an important discussion for your work.

**Limitations:**

The true main limitation of the proposed technique is mainly in its "degree of novelty", in the sense that similar ideas has been already proposed in the literature.

**Strengths And Weaknesses:**

The idea is interesting, in my opinion. The main weakness is that the paper is difficult to read in some part.
Moreover, some relationships and connections  with other methods in the literature should be discussed.

---

> ### Author Response · Authors · 2022-07-30
> **MCMC, multiple-try algorithm, etc.**
>
> The objective of the paper is not to propose "yet" a new MCMC algorithm but a method to reduce the bias of SNIS while keeping the variance explicitly controlled by a bound that is inversely proportional to the number of proposal samples.
>
> We repeat below the argument given to Reviewer 9JEd.
>
> We use the same "input" as the SNIS algorithm, in the sense that we draw samples from the proposal distribution and then compute the importance weights, which is exactly what is required when computing the "classical" SNIS estimator. In most applications of ML, this is where most of the computational effort lies (as we typically use complex sampling methods and computing the importance weights can be costly).
>
> We then propose to perform an additional "extra-randomization" to compute the final estimator (but, mind you, without drawing any new samples). The computational cost of the extra-randomization operation is very small. Thus, **the objective is not to define a new MCMC procedure, but a very simple algorithm to improve SNIS at the price of an extremely limited increase of computational complexity**.
>
> The extra-randomization is derived from the i-SIR algorithm with full recycling; however, we stress that **we do not need any new simulations or new calculations of importance weights**. All the samples and importance weights have been computed once and for all, and this is where 99% of the computational burden lies in the examples that we are dealing with.
>
> This extra-randomization allows us to tradeoff bias against variance, in the sense that we are able to significantly reduce bias at the cost of a (very modest) increase in variance. Thus, this is a "different" way to compute an estimator using the same input as the SNIS estimator, **and it is not a "new" MCMC algorithm**.
>
> The references you give are excellent, and we really like Martino's and Calderhead's papers; still, we are not completely sure whether these are relevant here. On the contrary, we are a bit afraid that this will reinforce the confusion—which we will clear up by rewriting the introduction of the paper—about the real objectives of our contribution.
>
> To sum up, the main contributions of our work are
> - the proposal of a **wrapper** that allows to reduce the bias of SNIS methods at minimal computational cost (for the cases of interest).
> - explicit bounds (Theorems 4 and 5) on bias and variance of the proposed estimator, showing the often dramatic bias reduction and the explicit control of the variance, respectively.
> - numerical results that clearly support our claims.

---

### Official Review · Reviewer_9JEd · 2022-07-12

**Rating:** 6
**Confidence:** 3
**Soundness:** 4 excellent
**Presentation:** 3 good
**Contribution:** 3 good

**Summary:**

This work analysis a well-known multiple-proposal MCMC kernel known as i-SIR. The main contributions are

1. Theorem 3, which bounds the rate at which the bias/error decays with the number of iterations or number of particles.
2. Theorems 4, which gives bias, mean-square error, and high-probability bounds for a Rao--Blackwellised estimator (9) which re-cycles (almost) all generated samples.

**Questions:**

In  Line 241, it is stated that "An approach very similar to BR-SNIS can be taken also in [the particle Gibbs] context". What does this refer to? Do you mean one could use a similar Rao--Blackwellisation or similar strategy of analysis?

**Limitations:**

Yes

**Strengths And Weaknesses:**

Strengths:

1. Novel bias/error bounds for the i-SIR algorithm and for the "Rao--Blackwellised" estimator from (9) (as well as a high-probability bound for the latter).

Weaknesses:

1. I was slightly disappointed after reading the paper because the title suggests a way to reduce the bias in self-normalised importance sampling (SNIS) but the analysed method is an MCMC algorithm. And the iterative nature of this MCMC algorithm may render it inapplicable in some settings in which "standard" SNIS is employed.

2. Theorems 1 and 2 are well known in the literature on particle MCMC methods. It might worth mentioning this somewhere on Page 3 to avoid the false impression that these are novel.

3. From reading the manuscript, it is not 100% clear to me whether the Rao--Blackwellised estimator in Equation 9 is claimed to be a novel contribution (Page 2 makes it sound as if it is). Just in case, it might be worth adding a reference to at least the first of the following sources where this Rao-Blackwellised estimator has been previously suggested (albeit without any theoretical analysis):

* Tjelmeland, H. (2004). Using all Metropolis–Hastings proposals to estimate mean values (No. NTNU-S-2004-4).

* Yang, S., Chen, Y., Bernton, E., & Liu, J. S. (2018). On parallelizable Markov chain Monte Carlo algorithms with waste-recycling. Statistics and Computing, 28(5), 1073-1081.

* Schwedes, T., & Calderhead, B. (2021, March). Rao-blackwellised parallel MCMC. In International Conference on Artificial Intelligence and Statistics (pp. 3448-3456). PMLR.

* Schwedes, T. (2019). Parallel Markov chain quasi-Monte Carlo methods. PhD thesis.




Typos et al:

- define $\mathbb{N}^*$
- $n$ -> $N$ in Theorem 4

---

> ### Author Response · Authors · 2022-07-30
> **I was slightly disappointed after reading the paper because the title suggests a way to reduce the bias in self-normalised importance sampling (SNIS) but the analysed method is an MCMC algorithm.**
>
> There is no reason for you to be disappointed, as **we do not propose an MCMC algorithm**, but a method to reduce the bias of SNIS while keeping the variance explicitely controlled by a bound that is inversely proportional to the number $M$ of samples from the proposal distribution.
>
> Note that we use the same "input" as the SNIS algorithm, in the sense that we draw $M$ samples from the proposal distribution and then compute the importance weights, which is exactly what is needed when computing the "classical" SNIS estimator. In most ML applications, this is where most of the computational burden lies (as we typically use complex sampling procedures and computing the importance weights might also be costly).
>
> We then propose to add, on the top of that, an "extra-randomization" to compute the final estimator (but, mind you, without drawing any new samples). The computational cost of the extra randomization is very small. The extra-randomization is derived from the i-SIR algorithm with full recycling, which does not require any new simulations or new calculations of importance weights; everything has been calculated once and for all, which corresponds to 99% of the calculations in the examples we have treated. This additional randomization allows us to trade-off bias against variance, in the sense that we are able to significantly reduce bias at the cost of a (very modest) increase in variance. **So you need not be disappointed, our procedure is indeed a "different" way to compute an estimator from the quantities computed for the SNIS estimator, and this is not a "new" MCMC**.

---

> > ### Comment · Reviewer_9JEd · 2022-08-07
> > **Terminology**
> >
> > I would still say that the analysed algorithm is more adequately characterised as an MCMC algorithm with multiple proposals (this would be in line with the literature) than as importance sampling. But I take your point that since you are specifically requiring that the proposals are "independent" proposals, i.e. that they do not depend on the previous state of the Markov chain, all the proposals can be generated in parallel at the start of the MCMC chain.

---

> ### Author Response · Authors · 2022-07-30
> **Theorems 1 and 2 are well known in the literature on particle MCMC methods. It might worth mentioning this somewhere on Page 3 to avoid the false impression that these are novel.**
>
> In fact, we mention in the introduction that the i-SIR algorithm goes back to the work of Tjelmeland [40], on which our work is based, rather than the (admittedly much better known and cited) particle Gibbs (PG) sampler, which is in some sense an extension of Tjemeland to the context of sequential Monte Carlo. Still, there is one small difference between our algorithm and that of Tjelmeland (and the PG sampler), in the sense that Tjelmeland keeps the index of the particle in the target distribution, while we instead keep the value of the selected sample. This small modification allows us to obtain a clean duality equality that does not appear in Tjemeland. As a result, this duality formula allows us to define a neat Gibbs procedure in two steps (whereas PG typically takes the form of a collapsed Gibbs sampler), which has very interesting theoretical properties (monotonicity, positivity, etc.). We agree, however, that this does not constitute the originality of our paper. Our goal was to allow the reader to follow our ideas without necessarily knowing all the arcana of PG, but we will change the text with this in mind.

---

> > ### Comment · Reviewer_9JEd · 2022-08-07
> > **Some additional context for Thm 1 and 2**
> >
> > Thank you for the detailed reply.
> >
> > I just want to point out that it is well known in the literature on particle Gibbs samplers that you don't need to keep the index of the reference path in the (extended) target distribution. That is, you can write down the induced Markov kernel without the need for resorting to "partially- collapsed Gibbs" sampling type arguments. For instance, see Section 4 in:
> >
> > Andrieu, C., Lee, A., & Vihola, M. (2018). Uniform ergodicity of the iterated conditional SMC and geometric ergodicity of particle Gibbs samplers. Bernoulli, 24(2), 842-872.

---

> > > ### Author Response · Authors · 2022-08-09
> > > **Thanks for your remark**
> > >
> > > Thanks a lot for this remark. We have added a reference to the work you mention at the point where we introduce the induced Markov kernel in question.

---

> ### Author Response · Authors · 2022-07-30
> **From reading the manuscript, it is not 100% clear to me whether the Rao--Blackwellised estimator in Equation 9 is claimed to be a novel contribution (Page 2 makes it sound as if it is). Just in case, it might be worth adding a reference to at least the first of the following sources where this Rao-Blackwellised estimator has been previously suggested (albeit without any theoretical analysis):**
>
> We are sorry for having given the impression of misquoting existing results. We wanted to give the logical "flow" of the construction, which led us to formulate Theorem 2, but we agree on that we should have been more careful in saying that this (Rao–Blackwellisation) result already appears in the work by Tjelmeland. In addition, we will of course add the other interesting references that you have indicated.

---

> ### Author Response · Authors · 2022-07-30
> **In Line 241, it is stated that "An approach very similar to BR-SNIS can be taken also in [the particle Gibbs] context". What does this refer to? Do you mean one could use a similar Rao--Blackwellisation or similar strategy of analysis?**
>
> Here we wanted to say that it is possible to extend the ideas and results of our paper to the sequential Monte Carlo and particle Gibbs frameworks, in order to design bias-reduced particle estimators of expectations under Feynman–Kac path distributions. Obviously, the derivations and the calculations of bias and variance will be considerably more complicated in that case.

---

> ### Author Response · Authors · 2022-07-30
> **Thanks for you comments … and please read our answers in the reverse order!**
>
> Thank you very much for your very pertinent comments (possibly with the exception of the first comment), which will help us improving the presentation of our results. We suggest you read our answers in reverse order, since we provide the same in the order of your questions.

---

### Author Response · Authors · 2022-08-01
**New revision**

We thank all the reviewers for the relevant commentaries. We have taken the commentaries into account, and we have uploaded a new revision with added references and an updated numeric section with better results and presentation.

---

### Author Response · Authors · 2022-08-05
**Update in numerics**

We have updated the numeric section with a comparison between our estimator and the zero bias estimator presented in [Middleton, L., Deligiannidis, G., Doucet, A., and Jacob, P. E. (2019). Unbiased Smoothing using Particle Independent Metropolis-Hastings. Proceedings of Machine Learning Research vol. 89. https://proceedings.mlr.press/v89/middleton19a.html]. Even though the two estimators have different goals (our goal is to provide a wrapper that allows to reduce the bias of SNIS methods at minimal computational cost not a zero bias estimator), it can be of interest to compare both of them in the case where there is a restriction in the total number of  available samples.

The table below displays approximate $95\\%$ intervals for both estimators under a strict budget of $8.192 \cdot 10^7$ samples for several configurations of each estimator in the Mixture of Gaussians example. The intervals were re-centered by the reference value.

|N|k|algorithm|center|lower bound|upper bound|interval length|
|    :----:   |    :----:   |    :----:   |    :----:   |    :----:   |    :----:   |    :----:   |
|129|512|BR-SNIS|0.0002|-0.0034|0.0038|0.0072|
|129|1024|BR-SNIS|-0.0001|-0.0037|0.0034|0.0071|
|257|512|BR-SNIS|0.0004|-0.0033|0.0040|0.0073|
|513|256|BR-SNIS|0.0013|-0.0025|0.0050|0.0075|
|65536|N/A|Unbiased-PIMH|-0.0013|-0.0052|0.0026|0.0078|
|32768|N/A|Unbiased-PIMH|-0.0020|-0.0061|0.0022|0.0084|
|16384|N/A|Unbiased-PIMH|-0.0008|-0.0051|0.0035|0.0086|
|8192|N/A|Unbiased-PIMH|-0.0020|-0.0067|0.0028|0.0095|
|2048|N/A|Unbiased-PIMH|0.0028|-0.0030|0.0085|0.0115|

More details are shown in the latest revision.

---

### Meta-Review · Area_Chair_79aJ · 2022-08-22

**Recommendation:** Accept
**Confidence:** Less certain

**Metareview:**

Importance sampling requires the knowledge of the normalization constant of the distribution to be sampled from. SNIS (Self-Normalized Importance Sampling) does not, but is biased. The authors study methods to reduce the bias in SNIS: BR-SNIS. They prove error bounds for this method (Theorems 3 and 4).

The reviewers agreed that BR-SNIS does a nice job in practice, and that the theoretical analysis is novel and sound [especially Reviewers 9JEd, 1FgZ].

However, they also all pointed out that the paper is overstating the novelty of BR-SNIS. Some identified BR-SNIS to the exising i-SIR/multiple proposal MCMC [9JEc]. Other reviewers wrote that they are not exactly the same but that BR-SNIS is strongly based on i-SIR and that this is not emphasized enough by the authors [nbwq, 1FgZ]. They all required that the author to make it clear that i-SIR is an existing method, that BR-SNIS is strongly based on it, to discuss the connection to multiple proposal MCMC thoroughly and to add many references. Overall, "the novelty in methodology seem to be not big" [hWyK], so the authors should make it clear that their main contribution is the analysis of the method (Theorems 3 and 4). One of them also required to clarify the novelty of the Rao-Blackwellised estimator [9JEd]. Finally, while they agree that Theorems 1 and 2 are necessary to understand BR-SNIS, it should also be stressed that these results are "classical results", on the contrary to Theorems 3 and 4 [9JEd].

The authors acknowledged this, and promised to fix the paper accordingly. During the discussion, the reviewers agreed that the theoretical analysis in the paper is interesting enough to justify its acceptation. I will therefore recommend to accept the paper. I will ask the authors to carefully include in the paper all the discussion on i-SIR and multiple proposal MCMC and on the novelty of BR-SNIS in general (and of course Rao-Blackwellised estimator // Theorems 1 and 2). There was also some potential computational problem pointed out by [hWyK], who was finally addressed by the authors reply: please implement the corresponding corrections in the paper if necessary.

**Award:**

No

---

### Decision · Program_Chairs · 2022-09-14

Accept